# Hide&Seek: Learning to Explain in an End-to-End Differentiable Network

**Tal Ellinson** [1]   **Hadi Mohasel Afshar** [1]   **Sally Cripps** [1]

## Abstract

Instance-wise feature selection is a valuable tool for interpreting labeled data and the predictions of black-box models. In contrast to global feature selection techniques, instance-wise methods dynamically identify important features for each instance. A growing number of methods learn a *selector*, which identifies important features, and a *predictor*, which uses these to make predictions. However, these pioneering methods face challenges including information leakage and lack of differentiability, which can slow training. In this paper, we present Hide&Seek, an end-to-end differentiable model for instance-wise feature selection. We jointly learn feature selection and prediction under a single objective without information leakage. Hide&Seek outperforms existing state-of-the-art models across a range of experiments and is fast to train. We achieve this by reformulating feature removal as a differentiable operation where instead of discretely removing features, we replace a proportion of each feature. Training is further stabilized via a parsimony-weight annealing framework.

## 1. Introduction and Context

Feature selection (FS) techniques help identify informative variables in input-output systems. They are used widely to interpret labeled datasets and explain the predictions of black-box models. For example, in genomics, FS has been used to analyze high-dimensional gene expression data and identify genes relevant to cancer diagnosis (Unger & Kather, 2024). Black-box models are now being used across domains from medical research (Kogan et al., 2020) to finance (Rudin & Shaposhnik, 2023) to the education sector (Levy et al., 2021). As public understanding of model risk rises,

decision makers increasingly require reasons behind model predictions. By identifying the important input variables, FS helps peer inside a black box and present a reason for a prediction, which can be sense-checked against a policy-maker's expertise. Beyond interpretability, it can simplify models and reduce overfitting by removing unnecessary features. Thus, FS is a key driver of scientific discovery, increased transparency and improved model performance.

There is an established literature on FS, which spans traditional (Liu & Setiono, 1996; Kohavi & John, 1997; Guyon & Elisseeff, 2003) and deep-learning (Covert et al., 2021; Xu & Yang, 2025) methods. Historically, techniques focused on global importance, identifying features with high predictive contribution across a whole dataset (Tibshirani, 1996; Louppe et al., 2013). However, there are limitations in such an approach. For example, diagnostic pathways can differ between patients and educational policies might be more or less effective for different student subpopulations. Recent years have given rise to algorithms for instance-wise feature selection (IWFS), which ascribe feature importance for each instance (e.g. patient or student) in a dataset.

A common approach of IWFS algorithms is to remove features from a prediction model and thereby quantify their importance (Covert et al., 2021). A pioneering method is LIME (Ribeiro et al., 2016), which perturbs features in the neighborhood of the input and fits a linear surrogate model to explain the prediction. SHAP (Lundberg & Lee, 2017) uses a game-theoretic framework to estimate the average marginal contribution of a feature across all feature permutations. These are both *additive feature attribution methods*, where predictions are decomposed into contributions from individual features.

A range of approaches exist and their classifications are not always mutually exclusive. Saliency methods include Integrated Gradients and DeepLIFT, which measure change in the output relative to a baseline for each input feature (Sundararajan et al., 2017; Shrikumar et al., 2017). Masking techniques include Dabkowski & Gal (2017) for image classification and Crabbé & Van Der Schaar (2021) for time-series analysis. Other work leverages attention mechanisms (Arik & Pfister, 2021; Choi et al., 2016) and causal framings (Schwab & Karlen, 2019). See Covert et al. (2021) for a useful taxonomy.

---

[1]Human Technology Institute; School of Mathematical and Physical Sciences, University of Technology Sydney, Australia. Correspondence to: Tal Ellinson <tal.ellinson@student.uts.edu.au>.

*Proceedings of the 43rd International Conference on Machine Learning*, Seoul, South Korea. PMLR 306, 2026. Copyright 2026 by the author(s).

*Table 1.* Comparison of similar instance-wise feature selection models. All four models use a selector–predictor framework.

|  | Hide&Seek | REAL-x | INVASE | L2X |
|---|---|---|---|---|
| Adaptive number of important features | ✓ | ✓ | ✓ | ✗ |
| End-to-end differentiable | ✓ | ✗ | ✗ | ✓ |
| Avoids information leakage | ✓ | ✓ | ✗ | ✗ |

**Related Work.** The class of algorithms most closely related to our work train a *selector*, which identifies important features, and a *predictor*, which uses the chosen features to make predictions. These methods include L2X (Chen et al., 2018), INVASE (Yoon et al., 2018) and REAL-x (Jethani et al., 2021).[1] Let $(\mathbf{X}, Y)$ denote the input features and corresponding output labels. L2X aims to maximize the mutual information between $\mathbf{X}_S$, a subset of important features, and $Y$. Conversely, INVASE and REAL-x seek to minimize the KL divergence between the distributions $p(Y \mid \mathbf{X})$ and $p(Y \mid \mathbf{X}_S)$. In practice, these methods approximate conditioning on $\mathbf{X}_S$ by using an ablated version of the full input vector.

In all three methods, the selector learns probabilities for feature inclusion, which are used to sample features sent to the predictor. This discrete sampling is a non-differentiable operation and presents a challenge that each technique handles separately. L2X uses a Gumbel–Softmax relaxation for subset sampling (Maddison et al., 2016; Jang et al., 2017). This makes their architecture end-to-end differentiable but introduces a key limitation: the number of important features must be decided in advance and must be the same for all instances. INVASE uses an actor–critic methodology from reinforcement learning (Peters & Schaal, 2008) to bypass the need for differentiability. This produces good results but makes the model slow to train. REAL-x employs REBAR gradients to approximate the discrete distribution with its continuous relaxation (Tucker et al., 2017).

**Information Leakage.** A challenge faced by many ablation-based explanation methods, including L2X (Chen et al., 2018) and INVASE (Yoon et al., 2018), is information leakage (Jethani et al., 2021). In jointly trained selector–predictor models, this issue can arise when unselected features are replaced with an ablated value, typically zero (Chen et al., 2018; Yoon et al., 2018; Jethani et al., 2021), or with a mean feature value (Imrie et al., 2022). The predictor may learn to exploit the resulting ablation pattern, rather than relying only on the values of the selected features. In this paper, we study this failure mode in the

---

[1]Referred to as amortized explanation methods (AEMs) in Jethani et al. (2021).

presence of *switch features*. A feature is a switch feature when its induced partition, rather than its exact value, determines the response. This can occur directly, or indirectly by governing the importance of other features.

As an example, consider a switch feature $X_1$, *temperature*, which influences $Y$, *cafe profit*, through the other features. When $X_1 < 20°\text{C}$, let $Y$ be a function of *coffee sales* and *hot chocolate sales*. When $X_1 \geq 20°\text{C}$, let $Y$ be a function of *coffee sales*, *iced coffee sales* and *iced tea sales*. In all instances, $X_1$ is important. However, a jointly trained selector and predictor network whose goal is to minimize the number of important features can encode $X_1 = 0$ as a signal for the partition $X_1 \geq 20°\text{C}$. This allows the joint system to ablate $X_1$ to 0 for all instances of warm (or cold) weather, while preserving prediction accuracy. The end result is that $X_1$ is not identified as important in a significant number of instances. Leakage can also occur when a switch feature impacts $Y$ directly. Consider an alternative scenario where a switch feature $X_1$, *temperature*, directly determines $Y$, *ice cream sales*. Say that when $X_1 \geq 25°\text{C}$, sales are made and when $X_1 < 25°\text{C}$, they are not. As above, information leakage can occur by ablating $X_1$ to 0 for all instances of warm (or cold) weather.

Jethani et al. (2021) were the first to identify some cases of information leakage arising from the joint training of selector–predictor networks. They address it with REAL-x by decoupling training: the predictor is first trained independently to predict $Y$ from *randomly* ablated feature vectors. The selector network is then optimized to provide a minimal subset of important features that preserves prediction accuracy. Because of this, REAL-x avoids information leakage. However, the disjoint training of the predictor network creates two challenges: training the full model is not end-to-end differentiable and the predictor must be sufficiently expressive to predict optimally under arbitrary subsets of features. This is difficult as the number of possible selected features grows exponentially with the dimensionality of the data. The challenges faced by these related methods are summarized in Table 1 and motivate our model.

We prove information leakage in Appendix A and derive a lower bound on the achievable misidentification rate in Corollary A.2. Appendix A.1 compares our treatment of information leakage to that in Jethani et al. (2021).

**Contributions.** In this paper, we make four contributions: (1) We introduce Hide&Seek, a novel method for instance-wise feature selection. The selector and predictor are jointly trained in an end-to-end differentiable framework. We achieve differentiability by replacing a proportion of each feature, instead of discretely removing a feature subset. The model is fast to train and performs strongly in a range of experiments.

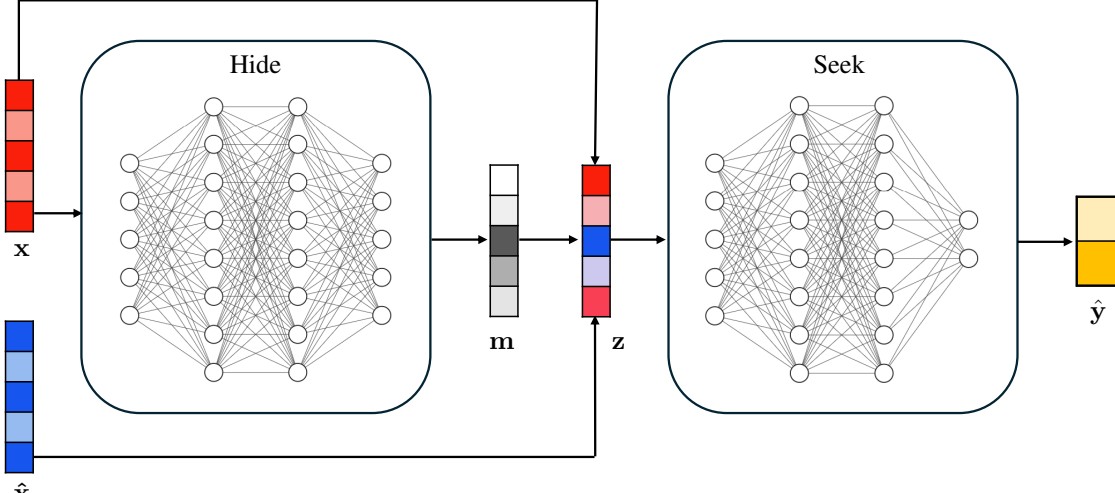

*Figure 1.* Hide&Seek architecture.

(2) We provide a mechanism for preventing information leakage while preserving joint training. Instead of replacing unselected features with a fixed ablation value, Hide&Seek uses random draws from the feature distribution. This prevents the predictor from exploiting the ablation as an extra source of information.

(3) We stabilize the joint training of the selector and predictor modules with a parsimony-weight annealing schedule, which markedly improves performance. The schedule allows the model to focus on predictive performance early in training and increase parsimony in the later stages.

(4) We study information leakage in a setting more general than that previously formalized (Jethani et al., 2021). We show that the typical loss function can be an ill-posed proxy for the intended loss and can favour omitting genuinely important features (Appendix A). We derive a lower bound on the achievable feature-importance misidentification rate (Corollary A.2), and our empirical results show that the observed misidentification rate is often close to this bound in practice.

This paper has the following structure. In Section 2, we formalize the mathematical foundation of our method. In Section 3, we describe the structure of our model. In Section 4, we evaluate Hide&Seek by comparing it against several benchmarks.

## 2. Problem Formulation

Consider a $d$-dimensional feature space $\mathcal{X} = \mathcal{X}_1 \times \cdots \times \mathcal{X}_d$ paired with an output space $\mathcal{Y}$. Let $\mathbf{x} := (x_1, \ldots, x_d)$ and $y$ be realizations of the random vector $\mathbf{X} := (X_1, \ldots, X_d) \in \mathcal{X}$ and random variable $Y \in \mathcal{Y}$, respectively. Let $D := \{1, \ldots, d\}$.

In instance-wise feature selection, the aim is to find a minimal subset $S \subseteq D$ per realization $\mathbf{x}$ that indexes the important features for predicting the label $y$. Let the reduced vector containing selected features be $\mathbf{x}_S$. We seek a minimum $|S|$ satisfying:

$$p(y \mid \mathbf{x}_S) = p(y \mid \mathbf{x}). \qquad (1)$$

However, modeling $p(y \mid \mathbf{x}_S)$ is non-trivial as the dimensionality of the reduced vector $\mathbf{x}_S$ changes between instances. A common strategy is to approximate $\mathbf{x}_S$ by a fixed-size vector $\mathbf{z} := (z_1, \ldots, z_d)$ in which

$$z_j = \begin{cases} x_j, & \text{if } j \in S, \\ \hat{x}_j, & \text{if } j \notin S. \end{cases} \qquad (2)$$

where $j \in \{1, \ldots, d\}$ and $\hat{\mathbf{x}} := (\hat{x}_1, \ldots, \hat{x}_d) \in \mathbb{R}^d$ represents uninformative replacement values such as $\mathbf{0}$ (Chen et al., 2018; Yoon et al., 2018; Tagaris & Stafylopatis, 2020; Jethani et al., 2021) or a vector of feature means (Imrie et al., 2022).

This approach introduces two problems that were discussed earlier. Specifically, removing features with eq. (2) requires discrete feature selection, which is non-differentiable, and ablation to fixed values, $\hat{x}_j$, can be exploited by a jointly trained selector and predictor network to produce information leakage. To address these limitations, we reformulate both the selection mechanism and the replacement strategy.

**Selection Mechanism.** To enable differentiable feature selection, we propose constructing $\mathbf{z}$ with a linear combination of the original signal, $x_j$, and a stochastic replacement value, $\hat{x}_j$:

$$z_j = m_j x_j + (1 - m_j)\hat{x}_j \qquad (3)$$

where $m_j \in [0,1]$ and a higher $m_j$ represents a higher likelihood that $j \in S$. In other words, instead of replacing a proportion of features, we replace a proportion of each feature.

Under this setting, the instance-wise feature selection task is equivalent to finding the most parsimonious real-valued mask vector $\mathbf{m} \in [0,1]^d$ such that

$$\mathbf{z} = \mathbf{m} \odot \mathbf{x} + (1 - \mathbf{m}) \odot \hat{\mathbf{x}}, \text{ and} \tag{4}$$

$$p(y \mid \mathbf{z}) = p(y \mid \mathbf{x}) \tag{5}$$

**Replacement Strategy.** If the modified signal, $\mathbf{z}$, is distributed according to the original feature distribution, i.e., $p(\mathbf{Z}) = p(\mathbf{X})$, then the predictor cannot distinguish between the original and replaced features and information leakage becomes impossible. Consider the case where $\mathbf{m} \in \{0,1\}^d$ is binary. In this context, $\hat{\mathbf{x}} = (\mathbf{x}_S, \hat{\mathbf{x}}_{\bar{S}})$, where $\bar{S} := D \setminus S$. No leakage is guaranteed when replacement values are drawn from the conditional distribution:[2]

$$\hat{\mathbf{x}}_{\bar{S}} \sim p(\mathbf{X}_{\bar{S}} \mid \mathbf{x}_S) \tag{6}$$

However, accurately modeling and sampling from this conditional distribution is non-trivial. In practice, we sample replacement values from the product of the marginal distributions, akin to unary Quantitative Input Influence (Datta et al., 2016):

$$\hat{\mathbf{x}} \sim \prod_{i \in D} p(X_i) \tag{7}$$

Note that we write $\hat{\mathbf{x}}$ rather than $\hat{\mathbf{x}}_{\bar{S}}$ because with our continuous relaxation, all features have a proportion selected (and unselected) via the continuous masks $\mathbf{m} \in [0,1]^d$.

Marginal sampling is efficient to implement and is significantly harder to exploit than fixed-value ablation. For leakage to occur in fixed-value ablation, the predictor network simply needs to learn the significance of a single ablated value. For leakage to occur under marginal sampling, the predictor network must first learn the true joint distribution of the data, and then recognize when stochastic replacement values are out-of-distribution. Our experiments show the absence of leakage under marginal replacement, including in highly correlated settings. In Appendix A.2, we also explore an alternative method to approximate the conditional distribution.

### 2.1. Optimization

As in Yoon et al. (2018), we relax (1) using the KL (Kullback-Leibler) divergence. Let $\mathcal{S} : \mathcal{X} \to 2^{\{1,\ldots,d\}}$ be a selector function that maps each input instance $\mathbf{x}$ to an instance-specific subset of selected feature indices. For notational convenience, we write $\mathbf{X}_S := \{X_i\}_{i \in \mathcal{S}(\mathbf{x})}$. Minimizing the KL divergence between $p(Y \mid \mathbf{X})$ and $p(Y \mid \mathbf{X}_S)$ is equivalent to minimizing the cross-entropy with respect to the selector function $\mathcal{S}$:

$$\min_{\mathcal{S}} \mathbb{E}_{\mathbf{X}} \left[ \mathrm{KL} \left( p(Y \mid \mathbf{X}) \,\|\, p(Y \mid \mathbf{X}_S) \right) \right]$$

$$= \min_{\mathcal{S}} \mathbb{E}_{\mathbf{X}} \mathbb{E}_{Y \mid \mathbf{X}} \left[ \log \frac{p(Y \mid \mathbf{X})}{p(Y \mid \mathbf{X}_S)} \right]$$

$$= \min_{\mathcal{S}} \mathbb{E}_{\mathbf{X}} \mathbb{E}_{Y \mid \mathbf{X}} \left[ \log p(Y \mid \mathbf{X}) - \log p(Y \mid \mathbf{X}_S) \right]$$

$$= \min_{\mathcal{S}} \underbrace{\mathbb{E}_{\mathbf{X}} \mathbb{E}_{Y \mid \mathbf{X}} [\log p(Y \mid \mathbf{X})]}_{\text{constant w.r.t. } \mathcal{S}}$$

$$+ \min_{\mathcal{S}} \mathbb{E}_{\mathbf{X}} \mathbb{E}_{Y \mid \mathbf{X}} [- \log p(Y \mid \mathbf{X}_S)]$$

$$\equiv \min_{\mathcal{S}} \mathbb{E}_{\mathbf{X}} \mathbb{E}_{Y \mid \mathbf{X}} [- \log p(Y \mid \mathbf{X}_S)]. \tag{8}$$

In practice, the true conditional distribution of $p(Y \mid \mathbf{X}_S)$ is unknown and is approximated by the model induced distribution $p_\theta(Y \mid \mathbf{X}_S)$. This allows us to define the following loss:

$$\mathcal{L}(\mathcal{S}, \theta) = \mathbb{E}_{(\mathbf{X},Y) \sim p} \left[ - \log p_\theta(Y \mid \mathbf{X}_S) \right]$$
$$+ \lambda \, \mathbb{E}_{\mathbf{X} \sim p_X} [\|\mathcal{S}(\mathbf{X})\|] \tag{9}$$

where $\| \cdot \|$ represents the $l_1$ norm, encouraging sparse outputs from $\mathcal{S}$. $\lambda$ is a regularization hyperparameter that balances minimizing the KL divergence against sparsity. For classification, the negative log-likelihood reduces to the standard cross-entropy loss.

## 3. Proposed Model

Take a dataset $\mathcal{D} = \{(\mathbf{x}^{(i)}, \mathbf{y}^{(i)})\}_{i=1}^n$ with $n$ i.i.d. samples drawn from the joint distribution $p(\mathbf{X}, \mathbf{Y})$, where $\mathbf{y} \in \{0,1\}^C$ represents the one-hot encoding of the target variable.[3] The Hide&Seek architecture consists of two feed-forward fully connected neural network modules: *Hide* and *Seek*.[4] As shown in Figure 1, *Hide* takes an input vector $\mathbf{x}$ and finds a continuous mask vector $\mathbf{m} \in [0,1]^d$, representing the importance of $d$ features. $\mathbf{z}$ is determined by (4) and (7) and *Seek* maps it to the predicted output $\hat{\mathbf{y}}$. With these choices, the loss function (9) becomes:

---

[2]See Appendix A.3 for full details.

[3]We focus on classification. In our experiments, $C = 2$, except in experiment 4.6 where $C = 4$. For regression, under a fixed-variance Gaussian likelihood, the cross-entropy term in the loss can be replaced with the mean-squared error.

[4]This naming convention was first introduced in Tagaris & Stafylopatis (2020).

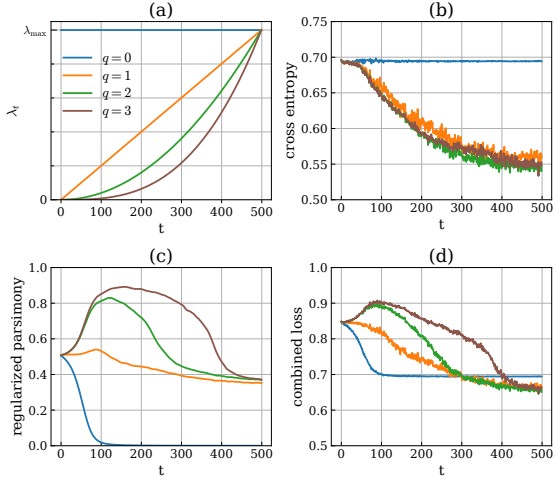

*Figure 2.* The annealing schedule for $\lambda_t$ over $t$ epochs, showing the impact of different choices of $q = \{0, 1, 2, 3\}$ in $\lambda_t = \left(\frac{t}{T}\right)^q \lambda_{\max}$ on the loss function (12). (a) The functional form of $\lambda_t$. (b) The cross-entropy term $-\sum_{c=1}^{C} y_c \log \hat{y}_c$ vs t. (c) The regularized parsimony term $\frac{\lambda_t}{d}\|\mathbf{m}\|_1$ vs t. (d) The combined loss $\ell(\boldsymbol{\alpha}, \boldsymbol{\beta}, t)$ vs t. The metrics in (b)-(d) are calculated on a hold-out validation set using the same $\lambda_{\max}$ and data as in the Syn4 experiment in section 4.1. See Appendix D.7 for other choices of $\lambda_{\max}$ and more detail.

$$\ell(\boldsymbol{\alpha}, \boldsymbol{\beta}) = \mathop{\mathbb{E}}_{\substack{(\mathbf{x},\mathbf{y}) \sim p(\mathbf{X},\mathbf{Y}) \\ \hat{\mathbf{x}} \sim \prod_{i \in D} p(X_i)}} \left[ -\sum_{c=1}^{C} y_c \log \hat{y}_c + \frac{\lambda}{d}\|\mathbf{m}\|_1 \right] \tag{10}$$

$$\begin{aligned} \text{where} \quad & \mathbf{m} = \text{Hide}_{\alpha}(\mathbf{x}) \\ & \mathbf{z} = \mathbf{m} \odot \mathbf{x} + (1 - \mathbf{m}) \odot \hat{\mathbf{x}} \\ & \hat{\mathbf{y}} = \text{Seek}_{\beta}(\mathbf{z}) \end{aligned}$$

In summary, feature importance is achieved without discrete sampling of features to retain or replace. Rather, using continuous masks, we replace a proportion of each feature with marginal noise, enabling end-to-end differentiability and the joint training of our network.

Further details on the model architecture are provided in Appendix B.[5]

### 3.1. Parsimony-Weight Annealing

Optimizing the parameters of a joint network to both minimize cross-entropy and increase parsimony presents a challenge. Specifically, placing excessive emphasis on parsimony early in training can cause the optimization to become trapped in poor local minima. To address this, we introduce an annealing schedule, in line with Bowman et al. (2016)

---

[5]Our code is available at https://github.com/talellinson/hide-and-seek-icml2026.

and Tagaris & Stafylopatis (2020). In our method, we progressively increase the weight on our parsimony term by quadratically growing the regularization parameter towards a set maximum value. Let $\lambda_{\max}$ be a fixed hyperparameter, and $t \in \{1, \ldots, T\}$ index the training over $T$ epochs. We replace $\lambda$ in (10) with $\lambda_t$ where:

$$\lambda_t = \left(\frac{t}{T}\right)^2 \lambda_{\max} \tag{11}$$

and the final loss becomes:

$$\ell(\boldsymbol{\alpha}, \boldsymbol{\beta}, t) = \mathop{\mathbb{E}}_{\substack{(\mathbf{x},\mathbf{y}) \sim p(\mathbf{X},\mathbf{Y}) \\ \hat{\mathbf{x}} \sim \prod_{i \in D} p(X_i)}} \left[ -\sum_{c=1}^{C} y_c \log \hat{y}_c + \frac{\lambda_t}{d}\|\mathbf{m}\|_1 \right] \tag{12}$$

As shown in Figure 2, this schedule allows the model to prioritize prediction accuracy early in training and focus on mask parsimony in later stages, leading to a lower final loss. Experimentally, this led to strong performance in the feature importance metrics. See Appendix D.7 for more detail.

## 4. Experiments

We present five analyses. The *Synthetic data* experiment evaluates Hide&Seek's ability to recover ground-truth feature importance against existing benchmarks. *Switch analysis* compares model performance in identifying switch-features, that is, features whose induced partition, rather than their exact feature value, affects the response. *Credit default data* measures Hide&Seek's ability to recover ground truth feature importance from highly correlated features, as well as comparing the predictive power of the model in semi-synthetic and real-world settings. The fourth experiment presents a Hide&Seek explanation for *MNIST* images. The final experiment, *Breast cancer subtype classification* applies Hide&Seek to genetic microarray data.

### 4.1. Synthetic Data

Our first experiment uses the same synthetic datasets as in Yoon et al. (2018), Chen et al. (2018) and Arik & Pfister (2021). The output $Y$ is sampled from a Bernoulli distribution, where $P(Y = 1|U) = \frac{1}{1+e^U}$. $U$ is a function of 11 Gaussian iid variables $X_1, ..., X_{11}$, where $X_j \sim \mathcal{N}(0, 1)$. There are six settings.

- **Syn1**: $U = X_1 X_2$

- **Syn2**: $U = \sum_{i=3}^{6} X_i^2 - 4$

- **Syn3**: $U = -10\sin(0.2X_7) + |X_8| + X_9 + e^{-X_{10}} - 2.4$

- **Syn4**: $U$: if $X_{11} < 0$, follow Syn1, else Syn2

*Table 2.* Performance of eight algorithms across six datasets. Each TPR and FDR metric is the median of 20 experiments. See Appendix C for boxplots.

| Model | Hide&Seek | | INVASE | | REAL-x | | SHAP | | LIME | | L2X | | RForest | | LASSO | |
|---|---|---|---|---|---|---|---|---|---|---|---|---|---|---|---|---|
| | TPR | FDR | TPR | FDR | TPR | FDR | TPR | FDR | TPR | FDR | TPR | FDR | TPR | FDR | TPR | FDR |
| Syn1 | **100** | **0** | **100** | **0** | **100** | 9 | 64 | 36 | 19 | 81 | 30 | 70 | **100** | **0** | 0 | 100 |
| Syn2 | **100** | **0** | **100** | **0** | **100** | 4 | 95 | 5 | **100** | **0** | 75 | 25 | **100** | **0** | 50 | 50 |
| Syn3 | 99 | **0** | 91 | **0** | 82 | 1 | 92 | 8 | 96 | 4 | 43 | 57 | **100** | **0** | 75 | 25 |
| Syn4 | **99** | 4 | 90 | **4** | 98 | 15 | 72 | 38 | 54 | 51 | 44 | 65 | 67 | 40 | 58 | 55 |
| Syn5 | **97** | 3 | 83 | **1** | 88 | 9 | 74 | 38 | 50 | 54 | 53 | 57 | 67 | 40 | 56 | 50 |
| Syn6 | **98** | 4 | 90 | 9 | 91 | **4** | 72 | 28 | 51 | 49 | 53 | 47 | 60 | 40 | 60 | 40 |

- **Syn5**: $U$: if $X_{11} < 0$, follow Syn1, else Syn3

- **Syn6**: $U$: if $X_{11} < 0$, follow Syn2, else Syn3

The goal is to identify the specific features used in generating $Y$. Syn1–3 represent global feature importance problems. Syn4–6 use a switch feature, $X_{11}$, to create instance-wise feature importance settings. We evaluate each algorithm's success using *True Positive Rate* (TPR) and *False Discovery Rate* (FDR), as in (Yoon et al., 2018).

We compare eight algorithms. Six are instance-wise algorithms: Hide&Seek, INVASE (Yoon et al., 2018), REAL-x (Jethani et al., 2021), SHAP (Lundberg & Lee, 2017), LIME (Ribeiro et al., 2016) and L2X (Chen et al., 2018). Two provide global feature importance: LASSO (Tibshirani, 1996) and Random Forest (Breiman, 2001). We train on 10,000 samples and test on 10,000 samples.

Hide&Seek outperforms the other models on instance-wise feature selection (IWFS) metrics, as shown in Table 2. Its end-to-end differentiability makes it easy to train, with model run times shown in Table 3. INVASE and REAL-x have good performance, but significantly longer training times due to more complex architectures. INVASE and L2X use different activation functions for different datasets (ReLU for Syn1-2 and SELU for Syn4–6) (Yoon et al., 2018; Chen et al., 2018). In Hide&Seek, the activation function (ReLU) is constant across all datasets.

Hide&Seek, INVASE and REAL-x have a natural mechanism for determining whether a feature is important, namely that its mask is greater than 0.5. This allows the number of important features to be automatically chosen for each instance. In all other methods, the number of important features, $k$, needs to be specified. In the experiments, $k$ is chosen based on the number of ground truth important features sought for each dataset. Specifically, $k = 2$ for Syn1, $k = 4$ for Syn2–3 and $k = 5$ for Syn4–6. This can overestimate the FDR for Syn4 and Syn5, which have 5 important features when $X_{11} \geq 0$ but only 3 important features when $X_{11} < 0$. To account for this, SHAP, LIME and L2X results for $k = 3$ and $k = 4$ are shown in Appendix D.4.

The distributions for the masks for each of the synthetic datasets are provided in Appendix D.1. Each model's predictive performance is shown in Appendix D.2. Hide&Seek shows a higher predictive performance than other selector–predictor models despite having $17\times$ (REAL-x, L2X) to $30\times$ (INVASE) fewer parameters.

### 4.2. Switch Analysis

Corollary A.2 shows that, under the ill-posed loss proxy, a switch feature can be ablated, and hence misidentified as unimportant, at an achievable rate at least as large as the probability of its largest partition cell. In Syn4–6, the switch feature is $X_{11}$. Since $X_{11} \sim \mathcal{N}(0, 1)$, the partition cells $X_{11} < 0$ and $X_{11} \geq 0$ have equal probability, and the corollary therefore gives an achievable misidentification rate of $50\%$. Intuitively, a jointly trained selector–predictor network that ablates unselected features to zero can use the artificial value $X_{11} = 0$ as a code for one of the two partition cells. The end result is that $X_{11}$ is not identified as important $50\%$ of the time, without degrading prediction accuracy.

To test for this information leak, we analyze each model's ability to correctly identify $X_{11}$ as an important feature. As in Section 4.1, we run each of Syn4–6 20 times. We then test how often each model correctly identifies the switch-feature, $X_{11}$. Across Syn4–6, REAL-x has perfect switch accuracy, due to its disjoint training of the predictor network. The median switch accuracy of Hide&Seek is also $100\%$ for Syn4–6, significantly higher than INVASE ($\approx 51\%$)

*Table 3.* Typical run times of selector–predictor models for the synthetic data, with IWFS metrics reported in table 2. Times include training (10,000 samples), prediction, and feature importance attribution (10,000 samples). See Appendix B.1 for hardware and Appendix D.5 for more detail.

| Method | Run time (hh:mm:ss) |
|---|---|
| L2X | 00:00:03 |
| Hide&Seek | 00:00:05 |
| REAL-x | 00:01:16 |
| INVASE | 01:18:52 |

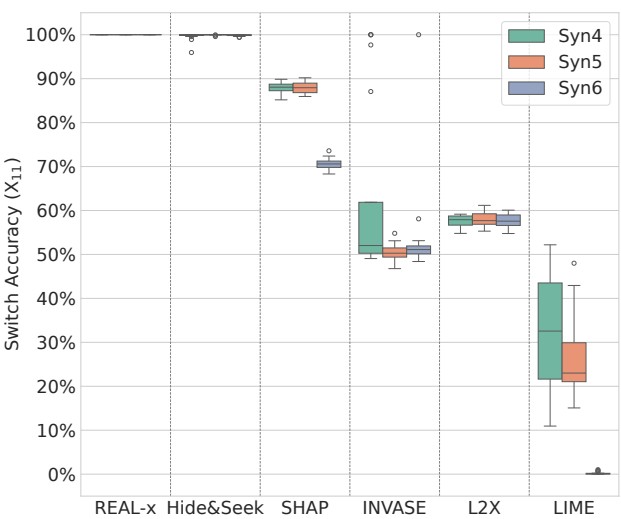

*Figure 3.* Switch accuracy. The percentage of instances where the switch-feature, $X_{11}$, was correctly identified as important. Each boxplot represents the distribution across 20 runs.

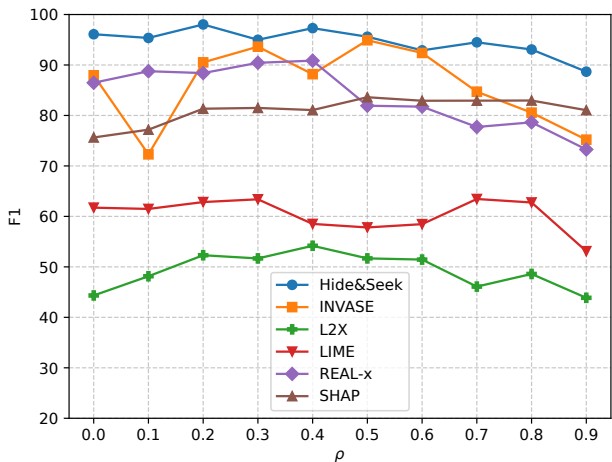

*Figure 4.* Model performance under the multicollinear setting. Each point represents the average IWFS F1 score across Syn1-6 for a given model and a given value of $\rho$. For these synthetic datasets, $\mathbf{X} \sim \mathcal{N}(\mathbf{0}, \Sigma)$, where $\Sigma_{i,j} = \rho$ for $i \neq j$ and $\Sigma_{i,j} = 1$ for $i = j$.

and L2X ($\approx 58\%$). These results in Figure 3 empirically confirm the benefit of our feature replacement strategy.

To assess the impact of this information leakage, we analyzed the INVASE results from the experiment in Section 4.1. We manually corrected failures to identify the switch feature, $X_{11}$, in Syn4–6 and recomputed the results. If $X_{11}$ were correctly identified, INVASE's TPR would rise to 99.9, 99.8, and 99.9 for Syn4, Syn5, and Syn6, respectively. Note that FDR would be unchanged as $X_{11}$ is always important. In other words, INVASE's TPR gap in Table 2 could be closed entirely by correctly identifying the switch feature.

### 4.3. Correlated Data

This experiment investigates the robustness of Hide&Seek under multicollinearity. We generate features and the target as in section 4.1, except that instead of drawing each feature independently from $X_j \sim \mathcal{N}(0, 1)$, we draw from a multivariate normal distribution $\mathbf{X} \sim \mathcal{N}(\mathbf{0}, \Sigma)$. The covariance matrix $\Sigma$ is defined using a constant pairwise correlation parameter $\rho$, such that $\Sigma_{i,j} = \rho$ for $i \neq j$ and $\Sigma_{i,j} = 1$ for $i = j$. Thus, $\rho$ controls feature dependence.

We ran each model on Syn1-6, under this correlated setting, for each of $\rho \in \{0, 0.1, 0.2, \ldots, 0.8, 0.9\}$. The results in Figure 4 show that Hide&Seek consistently outperforms the other models at instance-wise feature selection under multicollinearity.

Hide&Seek also maintained a switch accuracy greater than 99.4% (averaged across Syn4–6) for all choices of $\rho$. This indicates the absence of leakage even when the switch feature has high pairwise correlations. Switch feature accuracies

for each $\rho$ and model are shown in Appendix A.4.

### 4.4. Credit Default Data

In this experiment, we evaluate the strongest selector–predictor models against correlated real-world features in a semi-synthetic IWFS setting. We also report on their predictive power.

The *Default of Credit Card Clients* dataset contains 30,000 instances and 23 features describing demographic and financial attributes of credit card holders (Yeh & Lien, 2009). We generate a binary label synthetically to establish a ground-truth feature importance against which to evaluate our models, using the same functional forms for Syn4-Syn6 as in section 4.1. For $\{X_1, \ldots, X_{10}\}$ we use highly correlated features including customer bill and payment amounts across consecutive months and set *AGE* as the switch feature, $X_{11}$. Appendix F.1 presents the full list of features and the correlation matrix. Table 4 shows the instance-wise feature selection results.

While the primary goal of these models is to identify important instance-wise features, each model's predictive power may be of interest. A significantly low prediction rate could indicate an upper bound on a model's feature selection ability. We evaluated the predictive performance on the Credit Default data in each of the semi-synthetic settings and in predicting the true $Y$ values, which indicate whether or not a customer defaulted on their payment. The results are shown in Table 5.

*Table 4.* IWFS performance on highly correlated credit default data, where the target is generated as in Section 4.1. Each metric is the median of 20 experiments.

| Model | Hide&Seek | | | INVASE | | | REAL-X | | |
|---|---|---|---|---|---|---|---|---|---|
| | TPR | FDR | F1 | TPR | FDR | F1 | TPR | FDR | F1 |
| Syn1 | **100** | 1 | 99 | **100** | **0** | **100** | 100 | 22 | 86 |
| Syn2 | 67 | 26 | 67 | **100** | **0** | **100** | 49 | 13 | 59 |
| Syn3 | **100** | **0** | **100** | 98 | **0** | 99 | 63 | 11 | 69 |
| Syn4 | **87** | 34 | **74** | 58 | **31** | 57 | 63 | 35 | 62 |
| Syn5 | **96** | 28 | **81** | 80 | 28 | 74 | 79 | 41 | 66 |
| Syn6 | **87** | 39 | **71** | 73 | **31** | 62 | 48 | 38 | 52 |

*Table 5.* Prediction performance for highly correlated credit default data with synthetically generated labels (Syn1-6) and the credit default binary $Y$ labels (CDY) that indicate whether or not a customer defaulted on their payment. The metric is mean AUROC (± standard error) across 20 runs.

| | AUROC | | |
|---|---|---|---|
| Model | Hide&Seek | INVASE | REAL-X |
| Syn1 | **0.664 ± 0.002** | 0.601 ± 0.002 | 0.641 ± 0.003 |
| Syn2 | **0.885 ± 0.002** | 0.863 ± 0.002 | 0.878 ± 0.003 |
| Syn3 | **0.767 ± 0.002** | 0.752 ± 0.003 | 0.729 ± 0.005 |
| Syn4 | **0.828 ± 0.001** | 0.815 ± 0.002 | 0.809 ± 0.002 |
| Syn5 | **0.700 ± 0.002** | 0.675 ± 0.002 | 0.663 ± 0.003 |
| Syn6 | **0.867 ± 0.002** | 0.858 ± 0.001 | 0.845 ± 0.002 |
| CDY | 0.770 ± 0.002 | **0.771 ± 0.001** | 0.756 ± 0.002 |

## 4.5. MNIST

The MNIST dataset contains handwritten digits represented as grayscale images (LeCun et al., 2002). We train on 11,172 images and test on 1,397 images. As in (Chen et al., 2018), we select only the 3s and 8s, with the intention that the minor differences between the two digits could imply feature importance where the ground truth is unknown.

We compare five algorithms: Hide&Seek, INVASE, REAL-x, SHAP and LIME. To assign feature importance, SHAP and LIME first require a baseline model to interpret. For SHAP, we train XGBoost, a tree-based gradient boosting algorithm (Chen & Guestrin, 2016). For LIME we train an independent neural network. Hide&Seek, INVASE and REAL-x perform predictions by design and we use their predictor networks without adjusting any architecture. In all five cases, we achieve a classification accuracy of 99.1% or greater.

Each image contains $28 \times 28 = 784$ pixels. To identify the most important patches, we assign an importance score to each pixel using the given model. We then apply a $3 \times 3$ sliding window over the image, assigning each $3 \times 3$ patch an aggregated importance score. Figure 5 shows the four most important patches in each image, identified by each model.

These instance-wise explanation models are expected to adapt to each drawing and highlight important pixels in each context. Only Hide&Seek and REAL-x consistently

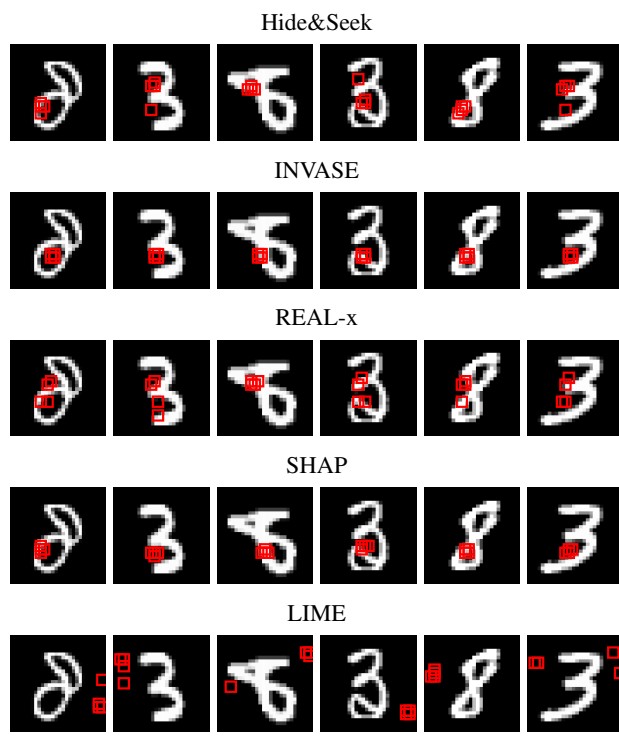

*Figure 5.* Important patches in MNIST data, as identified by each explanation method. Each 3x3 red square is one of the four most important patches in the image.

identify the 'left arcs' of each digit, which indicate an 8 when present and a 3 when absent. By contrast, LIME fails to identify important patches. INVASE and SHAP display less varied explainability.

## 4.6. Breast Cancer Subtype Classification

In this experiment, we use Hide&Seek to analyze breast cancer (BRCA) microarray data from The Cancer Genome Atlas (TCGA) where tumors are categorized into four different molecular subtypes (Berger et al., 2018). We select the same subset of 100 genes (out of 17,814) as in Covert et al. (2021) to reduce overfitting and enable comparison between our experiments. The dataset is relatively small, containing 572 instances. Using a test set of size 100, we rank the genes by mean mask size, thereby aggregating instance-wise importance into a global analysis. As shown in Table 6, the top two genes *ESR1* and *CCNB2* exhibit significantly higher importance than the remainder. Both *ESR1* (Robinson et al., 2013) and *CCNB2* (Shubbar et al., 2013) have established associations with BRCA and were also identified as important by Covert et al. (2021). We provide a comparison with other models, for which we updated the code of INVASE, REAL-x and L2X to handle classification with more than two classes. This experiment highlights the potential of Hide&Seek to identify biologically relevant

*Table 6.* Top 10 important genes across different explanation methods. Feature importance represents the mean mask size (Hide&Seek, INVASE, REAL-x), mean importance scores (SHAP, LIME), or mean selection frequency (L2X). Standard errors are provided in the appendix.

| Hide&Seek | | INVASE | | REAL-x | | SHAP | | L2X | | LIME | |
|---|---|---|---|---|---|---|---|---|---|---|---|
| Gene | Imp. | Gene | Imp. | Gene | Imp. | Gene | Imp. | Gene | Imp. | Gene | Imp. |
| ESR1 | 0.999 | CCNB2 | 0.723 | ESR1 | 0.969 | ESR1 | 0.695 | PENK | 0.640 | TUBB | 0.044 |
| CCNB2 | 0.951 | ESR1 | 0.711 | CCNB2 | 0.897 | CCNB2 | 0.402 | BIRC3 | 0.600 | HACE1 | 0.044 |
| STATH | 0.827 | ZNF775 | 0.680 | NUP210 | 0.833 | C6orf15 | 0.212 | TMEM52 | 0.590 | C6orf26 | 0.038 |
| C6orf26 | 0.804 | KLF3 | 0.619 | C6orf15 | 0.758 | ZNF385 | 0.126 | HPS4 | 0.590 | PENK | 0.033 |
| TUBB | 0.784 | C6orf15 | 0.610 | SLC25A3 | 0.754 | NUP210 | 0.121 | OTUD3 | 0.590 | ESR1 | 0.033 |
| C7 | 0.773 | TMSB10 | 0.597 | SPOCD1 | 0.606 | TMSB10 | 0.077 | CAPZB | 0.590 | C7 | 0.032 |
| NCAPH2 | 0.734 | NCAPH2 | 0.586 | C6orf26 | 0.593 | C6orf26 | 0.071 | C6orf26 | 0.580 | CCNB2 | 0.031 |
| UPK3B | 0.727 | C20orf111 | 0.570 | TUBB | 0.506 | C7 | 0.069 | STXBP1 | 0.580 | KIAA1949 | 0.029 |
| PARP1 | 0.710 | NUP210 | 0.562 | OR52E8 | 0.502 | HACE1 | 0.064 | ACLY | 0.580 | NCAPH2 | 0.029 |
| HACE1 | 0.680 | CAPZB | 0.553 | HACE1 | 0.498 | GPX2 | 0.062 | COL25A1 | 0.580 | OAS2 | 0.029 |

gene associations, including candidates beyond those previously reported in the literature. See Appendix F.3 for further details.

## 5. Conclusion

We introduced Hide&Seek, an end-to-end differentiable method for instance-wise feature selection. The method jointly trains a selector and predictor while avoiding a key failure mode: information leakage. Instead of ablating unselected features with a fixed value, Hide&Seek replaces them with random draws from the feature distribution. This prevents the predictor from using the ablation pattern itself as a source of information. Our use of continuous masks enables efficient joint optimization: rather than sampling a discrete subset of features, we replace a proportion of each feature. Our parsimony-weight annealing schedule stabilizes training by prioritizing predictive accuracy early and sparsity in later stages.

We also provided a theoretical analysis of information leakage in jointly trained ablation models. In particular, we showed that the usual training objective for ablation-based models can be an ill-posed proxy for the intended expected KL objective. This causes genuinely important features to be omitted without degrading predictive performance. We show that for switch features, which affect the response through their induced partition, this yields a lower bound on the achievable misidentification rate. Our experiments support this analysis: leakage-prone methods often fail to identify the switch feature at rates close to the theoretical bound, whereas Hide&Seek maintains high switch accuracy.

Under the idealized Hide&Seek construction: a binary mask and drawing replacement values conditioned on the selected features, information leakage is theoretically impossible. Our empirical results further show that Hide&Seek remains robust to information leakage when using a continuous relaxation of the mask and drawing replacement values from

the marginal feature distributions.

Across synthetic benchmarks, correlated semi-synthetic credit-default experiments, MNIST explanations, and genetic microarray data, Hide&Seek achieves strong instance-wise feature-selection performance while remaining fast to train. These results suggest that replacing fixed ablations with distributional feature replacement preserves the benefits of joint selector–predictor training without introducing an artificial information channel.

## Acknowledgements

This project was made possible by CSIRO's Next Generation Graduates program. We thank the reviewers for their thoughtful feedback.

## Author Contributions

H.A. conceived the main idea and developed the theoretical proofs. T.E. designed and implemented the model architecture and experiments. S.C. supervised the research, provided conceptual guidance and contributed to the interpretation of the results. All authors contributed to discussions and reviewed the final manuscript.

## Impact Statement

This paper presents work whose goal is to advance the field of Explainable AI (XAI). Improved interpretability of black-box models can support transparency and informed decision-making. However, incorrect or misleading explanations can introduce risk, which underscores the need for careful validation.

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

## A. On Feature Ablation and Information Leakage

Let the random vector $\mathbf{X} := (X_1, \ldots, X_d)$, defined on $\mathcal{X}$, represent the input features and let $Y$, defined on $\mathcal{Y}$, represent the response. Let $P^\star(\mathbf{X})$ and $P^\star(Y \mid \mathbf{X} = \mathbf{x})$ denote the empirical marginal distribution of $\mathbf{X}$ and conditional distribution of $Y$ induced by the observational data.

The ultimate objective of instance-wise feature selection (IWFS) setting is to find a *selector* function $\mathcal{S}(\mathbf{x})$ and and a *predictor* distribution $P_{\cdot|\mathbb{X}}(Y|\{X_k = x_k\}_{k \in \mathcal{S}(\mathbf{x})})$ such that the selector $\mathcal{S} : \mathcal{X} \to 2^{\{1, \ldots, d\}}$ maps each instance, $\mathbf{x}$, to a subset of $\{1, \ldots d\}$ indicating the indices of the important features and the pair $(\mathcal{S}, P_{\cdot|\mathbb{X}})$ minimizes the loss:

$$\ell_{\mathbb{X}}(\mathcal{S}, P_{\cdot|\mathbb{X}}) = \mathbb{E}_{\mathbf{x} \sim P^\star(\mathbf{X})} \Big[ \mathrm{KL}\big(P^\star(Y \mid \mathbf{X} = \mathbf{x}) \parallel P_{\cdot|\mathbb{X}}(Y \mid \{X_k = x_k\}_{k \in \mathcal{S}(\mathbf{x})})\big) + \lambda \|\mathcal{S}(\mathbf{x})\| \Big].$$

We assume that $\lambda > 0$ is chosen sufficiently small so that the minimiser of the intended loss $\ell_{\mathbb{X}}$ selects every important feature.

As mentioned in the main text, modelling this loss is challenging because it requires conditioning on arbitrary subsets of random variables. Ablation-based methods address this by converting $\{X_k = x_k\}_{k \in \mathcal{S}(\mathbf{x})}$ into a fixed-size ablated vector, $\mathbf{Z}$, via *feature ablation*.

That is, if $k \in \{1, \ldots, d\}$ and $k \notin \mathcal{S}(\mathbf{x})$, then the corresponding ablated element, $Z_k$, is replaced by a symbol, e.g., $*$, that is not in the original support, $\mathcal{X}_k$.[6] In many XAI methods, $* = 0$ (Zeiler & Fergus, 2014; Petsiuk et al., 2018; Yoon et al., 2018; Schwab & Karlen, 2019), though user-defined values are also common (Ribeiro et al., 2016; Dabkowski & Gal, 2017).

Ablating the features of an instance $\mathbf{X} = \mathbf{x}$ according to the selector $\mathcal{S}(\mathbf{x})$ is equivalent to constructing a random vector $\mathbf{Z}$ using a binary *ablation mask* $\mathcal{M} \in \{0, 1\}^d$:

$$\mathbf{Z} = \mathbf{x} \circledast \mathcal{M}(\mathcal{S}(\mathbf{x})) \text{ where } [\mathcal{M}(\mathcal{S}(\mathbf{x}))]_i = \begin{cases} 1, & \text{if } i \in \mathcal{S}(\mathbf{x}), \\ 0, & \text{otherwise.} \end{cases} \text{ and } [\mathbf{x} \circledast \mathbf{m}]_i = \begin{cases} x_i, & m_i = 1, \\ *, & m_i = 0. \end{cases}$$

These methods then minimise the following proxy loss:

$$\ell_{\mathbb{Z}}(\mathcal{S}, P_{\cdot|\mathbb{Z}}) := \mathbb{E}_{\mathbf{x} \sim P^\star(\mathbf{X})} \Big[ \mathrm{KL}\big(P^\star(Y \mid \mathbf{X} = \mathbf{x}) \parallel P_{\cdot|\mathbb{Z}}(Y \mid \mathbf{Z} = \mathbf{x} \circledast \mathcal{M}(\mathcal{S}(\mathbf{x})))\big) + \lambda \|(\mathcal{S}(\mathbf{x}))\| \Big].$$

The following theorem shows that $\ell_{\mathbb{Z}}(\mathcal{S}, P_{\cdot|\mathbb{Z}})$ can be an ill-posed proxy for $\ell_{\mathbb{X}}(\mathcal{S}, P_{\cdot|\mathbb{X}})$. In particular, in a broad class of scenarios, the selector and predictor that minimise the intended loss $\ell_{\mathbb{X}}(\mathcal{S}, P_{\cdot|\mathbb{X}})$ are not minimisers of the proxy loss $\ell_{\mathbb{Z}}(\mathcal{S}, P_{\cdot|\mathbb{Z}})$. Equivalently, there exist achievable selector–predictor pairs with lower proxy loss than the intended selector–predictor pair. As the theorem shows, such pairs exploit the ablation mask in an unintended way, allowing information to leak between the selector and predictor.

**Theorem A.1.** *Let a subspace, $\mathcal{A}_i \subseteq \mathcal{X}_i$, of a random variable $X_i \in \mathbf{X}$ be partitioned as follows: $\mathcal{A}_i := \bigsqcup_{k=1}^N A_{i,k}$ (where $N \geq 1$). Assume $Y$ depends on $X_i$ through its partition i.e., if the realization, $\mathbf{x}_{-i}$, of the other random variables, $\mathbf{X}_{-i} := \mathbf{X} \backslash \{X_i\}$, are given, $Y$ and $X_i$ are dependent but they are independent if the $X_i$'s partition is given:*

$$Y \not\perp\!\!\!\perp X_i \mid \mathbf{X}_{-i} = \mathbf{x}_{-i}, \qquad Y \perp\!\!\!\perp X_i \mid X_i \in A_{i,k}, \mathbf{X}_{-i} = \mathbf{x}_{-i}, \qquad \forall \mathbf{x}_{-i} \in \mathcal{X}_{-i}, \forall A_{i,k} \subset \mathcal{A}_i. \tag{13}$$

*Under this assumption, the loss function of the form:*

$$\ell_{\mathbb{Z}}(\mathcal{S}, P_{\cdot|\mathbb{Z}}) := \mathbb{E}_{\mathbf{x} \sim P^\star(\mathbf{X})} \Big[ KL\big(P^\star(Y \mid \mathbf{X} = \mathbf{x}) \parallel P_{\cdot|\mathbb{Z}}(Y \mid \mathbf{Z} = \mathbf{x} \circledast \mathcal{M}(\mathcal{S}(\mathbf{x})))\big) + \lambda \|(\mathcal{S}(\mathbf{x}))\| \Big]. \tag{14}$$

*is an ill-posed proxy for*

$$\ell_{\mathbb{X}}(\mathcal{S}, P_{\cdot|\mathbb{X}}) = \mathbb{E}_{\mathbf{x} \sim P^\star(\mathbf{X})} \Big[ KL\big(P^\star(Y \mid \mathbf{X} = \mathbf{x}) \parallel P_{\cdot|\mathbb{X}}(Y \mid \{X_k = x_k\}_{k \in \mathcal{S}(\mathbf{x})})\big) + \lambda \|\mathcal{S}(\mathbf{x})\| \Big]. \tag{15}$$

*In the sense that:*

$$\arg\min_{\mathcal{S}} \ell_{\mathbb{Z}}(\mathcal{S}, P_{\cdot|\mathbb{Z}}) \neq \arg\min_{\mathcal{S}} \ell_{\mathbb{X}}(\mathcal{S}, P_{\cdot|\mathbb{X}}). \tag{16}$$

---

[6] If $X_k$ is a continuous random variable and has a Lebesgue density, then the ablated features can be replaced by any fixed constant $c$, even if $c \in \mathbf{X}_k$, since the probability mass associated with any particular point is 0.

*Proof.* For any ablated vector $\mathbf{z}$, define $\mathcal{I}(\mathbf{z})$ as the set of feature indices that are not ablated:

$$\mathcal{I}(\mathbf{z}) := \{k \in \{1, \ldots, d\} \text{ s.t. } z_k \neq *\}. \tag{17}$$

It is easy to verify that, for any selector function $\mathcal{S}$, if $\mathbf{z} = \mathbf{x} \circledast \mathcal{M}(\mathcal{S}(\mathbf{x}))$ and $* \notin \mathcal{X}$, then

$$\mathcal{I}(\mathbf{z}) = \mathcal{S}(\mathbf{x}), \text{ and } \{z_k\}_{k \in \mathcal{I}(\mathbf{z})} = \{x_k\}_{k \in \mathcal{S}(\mathbf{x})}, \tag{18}$$

This follows because

$$
\begin{array}{l}
k \in \mathcal{S}(\mathbf{x}) \implies [\mathcal{M}(\mathbf{x})]_k = 1 \implies z_k = x_k \implies k \in \mathcal{I}(\mathbf{z}) \\
k \notin \mathcal{S}(\mathbf{x}) \implies [\mathcal{M}(\mathbf{x})]_k = 0 \implies z_k = * \implies k \notin \mathcal{I}(\mathbf{z})
\end{array}
\quad, \qquad \forall \mathbf{x} \in \mathcal{X}, \forall k \in \{1, \ldots, d\}.
$$

Let

$$(\mathcal{S}^*, P^*_{\cdot|\mathbb{X}}) = \arg \min_{(\mathcal{S}, P_{\cdot|\mathbb{X}})} \ell_{\mathbb{X}}(\mathcal{S}, P_{\cdot|\mathbb{X}}). \tag{19}$$

denote the intended selector–predictor pair, i.e., the pair that minimises the intended loss. We define $P^*_{\cdot|\mathbb{Z}}$ as the predictor on ablated vectors corresponding to the intended predictor $P^*_{\cdot|\mathbb{X}}$ :

$$P^*_{\cdot|\mathbb{Z}}(Y|\mathbf{Z} = \mathbf{z}) := P^*_{\cdot|\mathbb{X}}(Y \mid \{X_k = z_k\}_{k \in \mathcal{I}(\mathbf{z})}) \tag{20}$$

Therefore,

$$P^*_{\cdot|\mathbb{Z}}(Y \mid \mathbf{Z} = \mathbf{x} \circledast \mathcal{M}(\mathcal{S}(\mathbf{x}))) \overset{(20)}{:=} P^*_{\cdot|\mathbb{X}}(Y \mid \{X_k = z_k\}_{k \in \mathcal{I}(\mathbf{z})}) \overset{(18)}{=} P^*_{\cdot|\mathbb{X}}(Y \mid \{X_k = x_k\}_{k \in \mathcal{S}(\mathbf{x})}). \tag{21}$$

Substituting (21) into the proxy loss (14) gives

$$\ell_{\mathbb{Z}}(\mathcal{S}^*, P^*_{\cdot|\mathbb{Z}}) = \ell_{\mathbb{X}}(\mathcal{S}^*, P^*_{\cdot|\mathbb{X}}). \tag{22}$$

Thus, to prove the theorem, it is enough to construct another pair $(\mathcal{S}', P'_{\cdot|\mathbb{Z}})$ such that $\ell_{\mathbb{Z}}(\mathcal{S}', P'_{\cdot|\mathbb{Z}}) < \ell_{\mathbb{Z}}(\mathcal{S}^*, P^*_{\cdot|\mathbb{Z}})$.

Assuming that the premises (13) of the theorem hold, define

$$P'_{\cdot|\mathbb{Z}}(Y|\mathbf{Z} = \mathbf{z}) := \begin{cases} P^*_{\cdot|\mathbb{X}}(Y \mid \{X_k = z_k\}_{k \in \mathcal{I}(\mathbf{z})}, X_i \in A_{i,1}), & \text{if } z_i = *, \\ P^*_{\cdot|\mathbb{X}}(Y \mid \{X_k = z_k\}_{k \in \mathcal{I}(\mathbf{z})}), & \text{otherwise.} \end{cases} \tag{23}$$

and

$$\mathcal{S}'(\mathbf{x}) := \begin{cases} \mathcal{S}^*(\mathbf{x}) \backslash \{i\}, & \text{if } x_i \in A_{i,1}, \\ \mathcal{S}^*(\mathbf{x}), & \text{otherwise.} \end{cases} \tag{24}$$

Intuitively, $\mathcal{S}'$ acts as the intended selector $\mathcal{S}^*$, except that when the $i$-th feature lies in the partition cell $A_{i,1}$, this feature is not selected. The predictor $P'_{\cdot|\mathbb{Z}}$ also acts like the intended predictor, except that when the $i$-th entry of its ablated input is $*$, it interprets this as evidence that $X_i \in A_{i,1}$.

The dependence assumption, $Y \not\perp X_i \mid \mathbf{X}_{-i} = \mathbf{x}_{-i}$ and the choice of $\lambda$, entails that regardless of the realisation of $\mathbf{X}$, $X_i$ is selected by the intended selector $S^*$. Hence,

$$i \in \mathcal{S}^*(\mathbf{x}) \implies [\mathcal{M}(\mathcal{S}^*(\mathbf{x}))]_i = 1 \implies [\mathbf{x} \circledast \mathcal{M}(\mathcal{S}^*(\mathbf{x}))]_i = x_i \neq *, \qquad \forall \mathbf{x} \in \mathcal{X}, \forall i \in \mathcal{S}^*(\mathbf{x}). \tag{25}$$

Now consider two cases, First, suppose $x_i \notin A_{i,1}$. Then,

$$\mathcal{S}'(\mathbf{x}) \overset{(24)}{=} \mathcal{S}^*(\mathbf{x}) \tag{26}$$

Hence, if $\mathbf{z} = \mathbf{x} \circledast \mathcal{M}(\mathcal{S}'(\mathbf{x}))$, then

$$z_i \overset{(26)}{=} [\mathbf{x} \circledast \mathcal{M}(\mathcal{S}^*(\mathbf{x}))]_i \overset{(25)}{=} x_i \neq * \implies P'_{\cdot|\mathbb{Z}}(Y|\mathbf{Z} = \mathbf{z}) \overset{(23)}{=} P^*_{\cdot|\mathbb{X}}(Y \mid \{X_k = z_k\}_{k \in \mathcal{I}(\mathbf{z})}). \tag{27}$$

Thus (26) and (27) entail that on this region, the instance-wise loss contribution of $(\mathcal{S}', P'_{\cdot|\mathbb{Z}})$, is the same as that of $(\mathcal{S}^*, P^*_{\cdot|\mathbb{X}})$:

$$\ell_{\mathbb{Z}}(\mathcal{S}', P'_{\cdot|\mathbb{Z}}) = \ell_{\mathbb{X}}(\mathcal{S}^*, P^*_{\cdot|\mathbb{X}}), \qquad \forall \mathbf{x} \in \mathcal{X} \text{ s.t.} x_i \notin A_{i,1}. \tag{28}$$

Second suppose $\mathbf{x} \in A_{i,1}$. Then $\mathcal{S}'(\mathbf{x}) \overset{(24)}{=} \mathcal{S}^*(\mathbf{x})\backslash\{i\}$ which entails:

$$z_i = *, \text{ and } \mathcal{I}(\mathbf{z}) = \mathcal{S}^*(\mathbf{x})\backslash\{i\}. \tag{29}$$

By (29) and (23):

$$P'_{\cdot|\mathbb{Z}}(Y|\mathbf{Z} = \mathbf{z}) = P^*_{\cdot|\mathbb{X}}(Y \mid \{X_k = z_k\}_{k \in \mathcal{S}^*(\mathbf{x})\backslash\{i\}}, X_i \in A_{i,1}) = P^*_{\cdot|\mathbb{X}}(Y \mid \{X_k = z_k\}_{k \in \mathcal{S}^*(\mathbf{x})}),$$

in which the last equality holds by the theorem's assumption that $Y$ depends on $X_i$ through its partition.

Therefore, for all $\mathbf{x} \in \mathcal{X}$ such that $x_i \in A_{i,1}$, the divergence terms in (15) and (14) are equal but $\|\mathcal{S}'(\mathbf{x})\| = \|\mathcal{S}^*(\mathbf{x})\backslash\{i\}\| = \|\mathcal{S}^*(\mathbf{x})\| - 1$. Thus, on this region, the instance-wise loss contribution of $(\mathcal{S}', P'_{\cdot|\mathbb{Z}})$ is strictly smaller than that of $(\mathcal{S}^*, P^*_{\cdot|\mathbb{X}})$, while as we showed by (28), on the complement region $x_i \notin A_{i,1}$, the instance-wise loss contributions of $(\mathcal{S}', P'_{\cdot|\mathbb{Z}})$ and $(\mathcal{S}^*, P_{\cdot|\mathbb{X}})$ are equal. Therefore

$$\ell_{\mathbb{Z}}(\mathcal{S}', P'_{\cdot|\mathbb{Z}}) < \ell_{\mathbb{Z}}(\mathcal{S}^*, P^*_{\cdot|\mathbb{Z}}) \overset{(22)}{=} \ell_{\mathbb{X}}(\mathcal{S}^*, P^*_{\cdot|\mathbb{X}}).$$

which completes the proof. □

**Corollary A.2** (Lower bound on the achievable feature-importance misidentification rate). *Suppose the support $\mathcal{X}_i$ of a random variable $X_i \in \mathbf{X}$ be partitioned as $\mathcal{X}_i := \bigsqcup_{k=1}^{N} A_{i,k}$ and suppose $Y$ depends on $X_i$ only through its partition, in the sense of (13). Then a lower bound on the achievable rate at which $X_i$ is ablated, and hence misidentified as unimportant, is $P^\star(X_i \in A_{i,\max})$ where*

$$A_{i,\max} \in \arg \max_{A_{i,k} \in \{A_{i,1}, \ldots, A_{i,N}\}} P^\star(X_i \in A_{i,k}).$$

*Proof.* In Theorem A.1, any partition cell $A_{i,k}$ of $\mathcal{A}_i = \mathcal{X}_i$ can be chosen to play the role of $A_{i,1}$. For any such choice, the constructed leakage solution ablates $X_i$ whenever $X_i \in A_{i,k}$. Therefore, choosing the cell $A_{i,\max}$ yields an achievable misidentification rate of $P^\star(X_i \in A_{i,\max})$. □

**Remark.** Corollary A.2 is an existence result. It does not imply that every selector–predictor pair trained to minimize the proxy loss will attain the stated misidentification rate. This is because the training procedure may fail to discover the leakage strategy constructed in the proof, for example, by converging to a local minimum. Nor does the corollary imply that the stated rate $P^\star(X_i \in A_{i,\max})$ is the largest possible misidentification rate: other leakage strategies may achieve lower proxy loss while ablating $X_i$ on a larger subset of the input space. Rather, the corollary guarantees that there exists a selector–predictor pair whose proxy loss is lower than that of the intended selector–predictor pair and whose misidentification rate is $P^\star(X_i \in A_{i,\max})$. Thus, this quantity provides a lower bound on the largest misidentification rate achievable through information leakage.

## A.1. Comparison to Jethani et al. (2021)

Jethani et al. (2021) were the first to identify information leakage in jointly trained selector–predictor networks, which they refer to as joint amortized explanation methods (JAMs). They proved this problem in two scenarios. First, in a classification task where the number of possible outputs does not exceed the number of input features, the selector can encode the class label through the binary ablation mask, for instance, via a one-hot vector. This allows the predictor to decode the output from the mask rather than from genuinely informative selected features. The second scenario is when the output is generated by a tree-structured process whose leaves use distinct predictive feature sets. Here, the selected leaf-specific feature set can reveal the active branch. As a result, non-leaf (control-flow) features can be omitted while preserving the predictive likelihood, and the sparsity penalty then favors their omission.

In this paper, we study the JAM information leakage problem in a setting where the output $Y$ may depend on a feature $X$ through the partition of the input space induced by that feature. As a simple example, consider a binary response whose probability changes according to the sign of $X$, e.g., $P(Y = 1 \mid X) = p_-$ if $X < 0$ and $P(Y = 1 \mid X) = p_+$ if $X \geq 0$, with $p_- \neq p_+$. In this case, the relevant information is whether $X \in (-\infty, 0)$ or $X \in [0, +\infty)$, rather than the exact value of $X$. We refer to such variables as *switch features* throughout. This setting is more general than the scenarios studied by

Jethani et al. (2021) and covers cases where the output is either real-valued or discrete, corresponding to regression and classification tasks, respectively. There is no restriction on the number of classification labels, and if the output is generated by a tree-structured process, there is no restriction on the feature sets used in different leaves. In Section A, we show that, in this relaxed setting, JAM algorithms such as L2X and INVASE can still suffer from information leakage. Specifically, we show that the loss function used by such algorithms is an ill-posed proxy for the expected KL loss that they aim to minimize, in the sense that the selected features that minimize the two loss functions are not the same.

### A.2. Sampling $\hat{x}$ Using Model-X Knockoffs

As outlined in the paper, leakage can occur when a predictor network learns extra information from its masked inputs. A jointly trained selector–predictor network can exploit this knowledge by masking important features while retaining prediction accuracy, for example with ablation to 0. In Hide&Seek, we draw our replacement values from the product of the marginals, per eq. (30). Our results in Section 4.2, Section 4.3 and Section A.4 demonstrate the absence of leakage in this setting. Nonetheless, as outlined in Section 2, a stronger theoretical guarantee against leakage can be achieved by sampling replacement values from eq. (31). We present an alternative method using Model-X Knockoffs to more closely approximate this sampling.

$$\hat{\mathbf{x}} \sim \prod_{i \in D} p(X_i) \tag{30}$$

$$\hat{\mathbf{x}}_{\bar{S}} \sim p(\mathbf{X}_{\bar{S}} \mid \mathbf{x}_S) \tag{31}$$

**Model-X Knockoffs.** A random vector $\tilde{X} \in \mathbb{R}^p$ is a Model-X knockoff of $X$ constructed to satisfy the following two properties (Candès et al., 2018):

(a) $(X, \tilde{X})_{\text{swap}(S)} \overset{d}{=} (X, \tilde{X})$ for any subset $S \subset \{1, \dots, p\}$

where $(\cdot)_{\text{swap}(S)}$ swaps the entries $X_j$ and $\tilde{X}_j$ for each $j \in S$.

(b) If there is a response variable $Y$, then $\tilde{X} \perp\!\!\!\perp Y \mid X$

In our implementation, we generate Model-X knockoffs (Romano et al., 2020) using the package provided at `https://web.stanford.edu/group/candes/deep-knockoffs/`. The results for Experiment 4.1 are shown in Figure 6a and the results for Experiment 4.3 are shown in Figure 6b. These results show that marginal sampling, which was used in the paper, outperforms knockoff sampling.

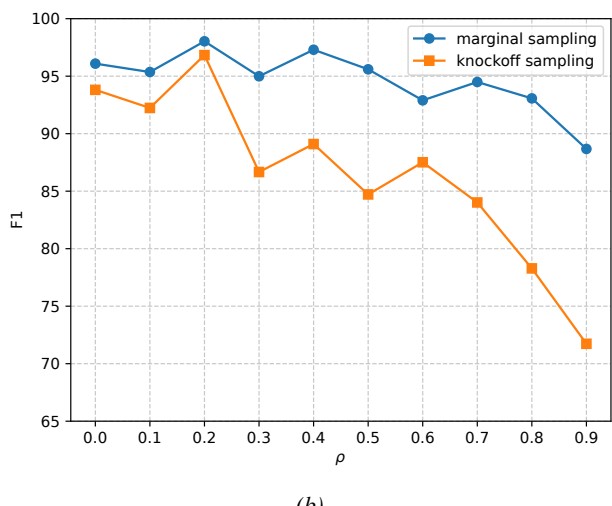

| Model | Marginal sampling | | Knockoff sampling | |
|---|---|---|---|---|
| | TPR | FDR | TPR | FDR |
| Syn1 | **100** | **0** | 99 | **0** |
| Syn2 | **100** | **0** | **100** | **0** |
| Syn3 | **99** | **0** | 98 | **0** |
| Syn4 | **99** | **4** | **99** | **4** |
| Syn5 | **97** | **3** | 96 | **3** |
| Syn6 | **98** | **4** | 89 | **4** |

*(a)*

*(b)*

*Figure 6.* Comparison of alternative methods for sampling $\hat{x}$ in the Hide&Seek architecture, showing that marginal sampling outperforms knockoff sampling. (a) can be directly compared to Table 2 in the paper and (b) can be directly compared to fig. 4.

Note that while the exchangeability property $(X, \tilde{X})_{\text{swap}(S)} \stackrel{d}{=} (X, \tilde{X})$ is guaranteed for any fixed subset $S$, this invariance does not hold in our context. This is because the selector–predictor architecture determines the masked subset using the selector network, meaning the chosen features are a function of the data itself, $S = f(X)$. Consequently, $(X, \tilde{X})_{\text{swap}(f(X))} \stackrel{d}{\neq} (X, \tilde{X})$.

### A.3. Preventing Information Leakage

Let us recall the following:

- $\mathbf{X} := (X_1, \ldots, X_d) \in \mathcal{X}$,

- $D := \{1, \ldots, d\}$,

- $S \subseteq D$   the indices of the selected features,

- $\bar{S} := D \setminus S$   the indices of the unselected features

- $\mathbf{z} = \mathbf{m} \odot \mathbf{x} + (1 - \mathbf{m}) \odot \hat{\mathbf{x}}$

In Section 2 we proposed that information leakage is prevented when replacement values are drawn from $\hat{\mathbf{x}}_{\bar{S}} \sim p(\mathbf{X}_{\bar{S}} \mid \mathbf{x}_S)$. The justification is as follows:

If the modified signal, $\mathbf{z}$, received by the predictor network is distributed according to the original feature distribution, i.e. $p(\mathbf{Z}) = p(\mathbf{X})$, then there is no way for the predictor to differentiate between the two and the information leakage becomes impossible. While, in our paper, we use a continuous relaxation where $\mathbf{m} \in [0, 1]^d$, let us consider the binary condition where $\mathbf{m} \in \{0, 1\}^d$.

In this setting, let $\mathbf{Z} := (\mathbf{X}_S, \hat{\mathbf{X}}_{\bar{S}})$. Thus, $p(\mathbf{Z}) = p(\mathbf{X}_S, \hat{\mathbf{X}}_{\bar{S}}) = p(\mathbf{X}_S)p(\hat{\mathbf{X}}_{\bar{S}} \mid \mathbf{X}_S)$. This distribution is equal to $p(\mathbf{X}) = p(\mathbf{X}_S, \mathbf{X}_{\bar{S}})$ if and only if $p(\hat{\mathbf{X}}_{\bar{S}} \mid \mathbf{X}_S) = p(\mathbf{X}_{\bar{S}} \mid \mathbf{X}_S)$. Thus to prevent any possible leakage $\hat{\mathbf{x}}$ should ideally be drawn from the conditional distribution: $\hat{\mathbf{x}}_{\bar{S}} \sim p(\mathbf{X}_{\bar{S}} \mid \mathbf{x}_S)$.

### A.4. Correlated Synthetic Data

Table 7 shows the switch feature accuracy for each of the models and $\rho$ values in the Section 4.3 experiment. It demonstrates that even with marginal sampling per eq. (7), the Seek module in Hide&Seek was unable to learn that replacement values for the switch feature were out-of-distribution.

*Table 7.* Switch accuracy (identifying $X_{11}$ as important) across $\rho$ and models. Each value is the mean switch accuracy across Syn4, Syn5 and Syn6 for a single run of each. Note that some variability in results is expected, as indicated by the boxplots in Figure 3.

| $\rho$ | Hide&Seek | INVASE | L2X | LIME | REAL-x | SHAP |
|---|---|---|---|---|---|---|
| 0.0 | 0.998 | 0.547 | 0.577 | 0.179 | 1.000 | 0.813 |
| 0.1 | 0.999 | 0.059 | 0.557 | 0.156 | 1.000 | 0.824 |
| 0.2 | 0.996 | 0.720 | 0.578 | 0.196 | 1.000 | 0.850 |
| 0.3 | 0.999 | 0.561 | 0.547 | 0.285 | 1.000 | 0.815 |
| 0.4 | 1.000 | 0.695 | 0.613 | 0.210 | 1.000 | 0.802 |
| 0.5 | 0.999 | 0.788 | 0.571 | 0.339 | 1.000 | 0.820 |
| 0.6 | 1.000 | 0.747 | 0.563 | 0.372 | 1.000 | 0.841 |
| 0.7 | 1.000 | 0.688 | 0.554 | 0.675 | 0.967 | 0.901 |
| 0.8 | 1.000 | 0.786 | 0.538 | 0.936 | 0.992 | 0.906 |
| 0.9 | 0.994 | 0.844 | 0.555 | 0.875 | 0.993 | 0.967 |

## B. Model infrastructure

**Hide&Seek**. Hide&Seek consists of two fully connected, feed-forward neural networks with ReLU activation functions. The last layer activation function of *Hide* is an element-wise sigmoid to ensure that the mask $\mathbf{m} \in [0,1]^d$, while the last layer of *Seek* uses a softmax activation. Each network has two hidden layers with ReLU activation functions and each hidden layer has 32 dimensions. The model is trained in 500 epochs without batching. We use the Adam optimizer (learning rate $= 0.001$), and model weights are initialized using the default PyTorch setting. The implementation is based on PyTorch v2.7.1 with CUDA 12.8. At each epoch, the training data columns are internally shuffled with replacement to create a dataset from which to draw $\hat{x}_j$ values. $\lambda_{\max} = 0.3$ for the synthetic data in Section 4.1. Our code is available at https://github.com/talellinson/hide-and-seek-icml2026.

**INVASE**. The implementation of INVASE uses the code in `https://github.com/iclr2018invase/INVASE`. Specifically, the selector (actor) network has two hidden layers, each with 100 dimensions. The predictor (critic) network has two hidden layers, each with 200 dimensions. The number of training epochs is 10,000, the batch size is 1,000 and $\lambda = 0.1$ for the synthetic data in Section 4.1.

**REAL-x**. The implementation of REAL-x uses the code in `https://github.com/rajesh-lab/realx`. Specifically, the selector network has two hidden layers, each with 100 dimensions. The predictor network has two hidden layers, each with 200 dimensions. The number of training epochs is 500, the batch size is 1,000 and $\lambda = 0.15$ for the synthetic data in Section 4.1.

**L2X**. The implementation of L2X uses the code in `https://github.com/Jianbo-Lab/L2X/tree/master`. Like INVASE, there are two networks, each with two hidden layers. Each hidden layer of the first network has 100 dimensions and each hidden layer of the second network has 200 dimensions.

**LIME**. The implementation of LIME uses the code in `https://github.com/marcotcr/lime/tree/master`. The baseline models for our Synthetic and MNIST data can be found in our repository.

**SHAP**. The implementation of SHAP uses the code in `https://github.com/shap/shap`. We explored two implementations of the SHAP package: KernelExplainer and TreeExplainer. Kernel SHAP uses weighted linear regression, similarly to LIME (Lundberg & Lee, 2017) and can be run on neural networks. Tree SHAP is a fast, tree-based algorithm that works with ensembles of trees. Tree SHAP performed better on our synthetic data, so we used it with a base XGBoost predictor (Chen & Guestrin, 2016). This combination explains the fast run time in Appendix D.5. The XGBoost model uses the code in `https://pypi.org/project/xgboost/`. The hyperparameters were chosen after tuning and are: {'objective': 'binary:logistic', 'eval_metric': 'logloss', 'max_depth': 5, 'eta': 0.1, 'colsample_bytree': 0.9, 'num_boost_round': 100} in Section 4.1.

**RForest**. The implementation of RForest uses the code in `https://scikit-learn.org/stable/modules/generated/sklearn.ensemble.RandomForestClassifier.html`. The hyperparameters are: {criterion='gini', n_estimators=100, max_depth=5}.

**LASSO**. The implementation of LASSO uses the code in `https://scikit-learn.org/stable/modules/generated/sklearn.linear_model.LogisticRegression.html`. We use logistic regression with an $L_1$ penalty.

### B.1. Hardware

Experiments were conducted on the following hardware:

- AMD EPYC 9354P 3.25GHz 32 cores 256MB L3 Cache (Max Turbo Freq. 3.75GHz)

- 192GB 4800MHz ECC DDR5-RAM (Twelve Channel)

- 1.92TB NVMe SSD Drive and 1.92TB NVMe SSD Drive

- 2x NVIDIA L4 (7,680 Cores, 240 Tensor Cores, 24GB Memory) GPUs

## C. Metric calculations

**Explaining the TPR, FDR and F1 metrics.** To compute the TPR, FDR and F1 values across a dataset, we first calculated the TPR, FDR and F1 score for each input instance (e.g. row in tabular data) using eq. (32) and eq. (33). The mean was then taken across the entire dataset. Where specified, each experiment was run 20 times using different seeds, with the median values reported in the tables. The full distributions for Section 4.1 are shown in Figure 7 as boxplots.

$$\text{TPR} = \frac{\text{true positives}}{\text{true positives} + \text{false negatives}} \qquad \text{FDR} = \frac{\text{false positives}}{\text{true positives} + \text{false positives}} \qquad (32)$$

$$\text{F1 score} = 2 \cdot \frac{\text{Precision} \cdot \text{Recall}}{\text{Precision} + \text{Recall}} \qquad (33)$$

where Recall $=$ TPR and Precision $= 1 - $ FDR.

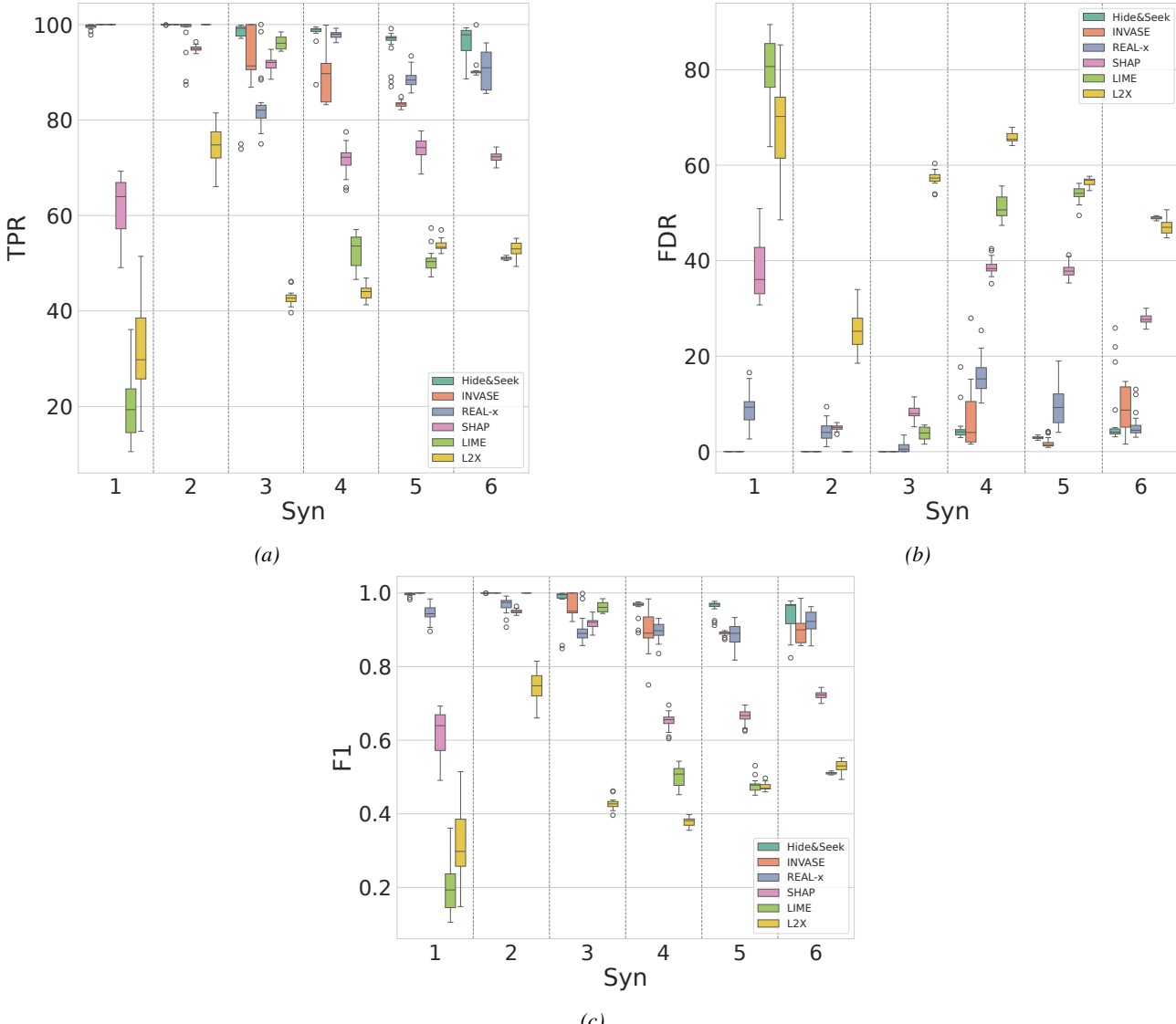

*(a)*

*(b)*

*(c)*

*Figure 7.* Distributions of (a) mean TPR, (b) mean FDR, and (c) mean F1 scores for feature identification across Syn1-6. Each boxplot shows the distribution across 20 experiments. The medians of the TPR and FDR boxplots are reported in Table 2.

# D. Further analyses

Section D contains analyses and experiments relating to the synthetic data experiment 4.1.

## D.1. Mask Distributions

Figure 8 shows the learned mask distributions for the six synthetic experiments in Section 4.1. Note the close alignment with the data-generating rules defined in Section 4.1.

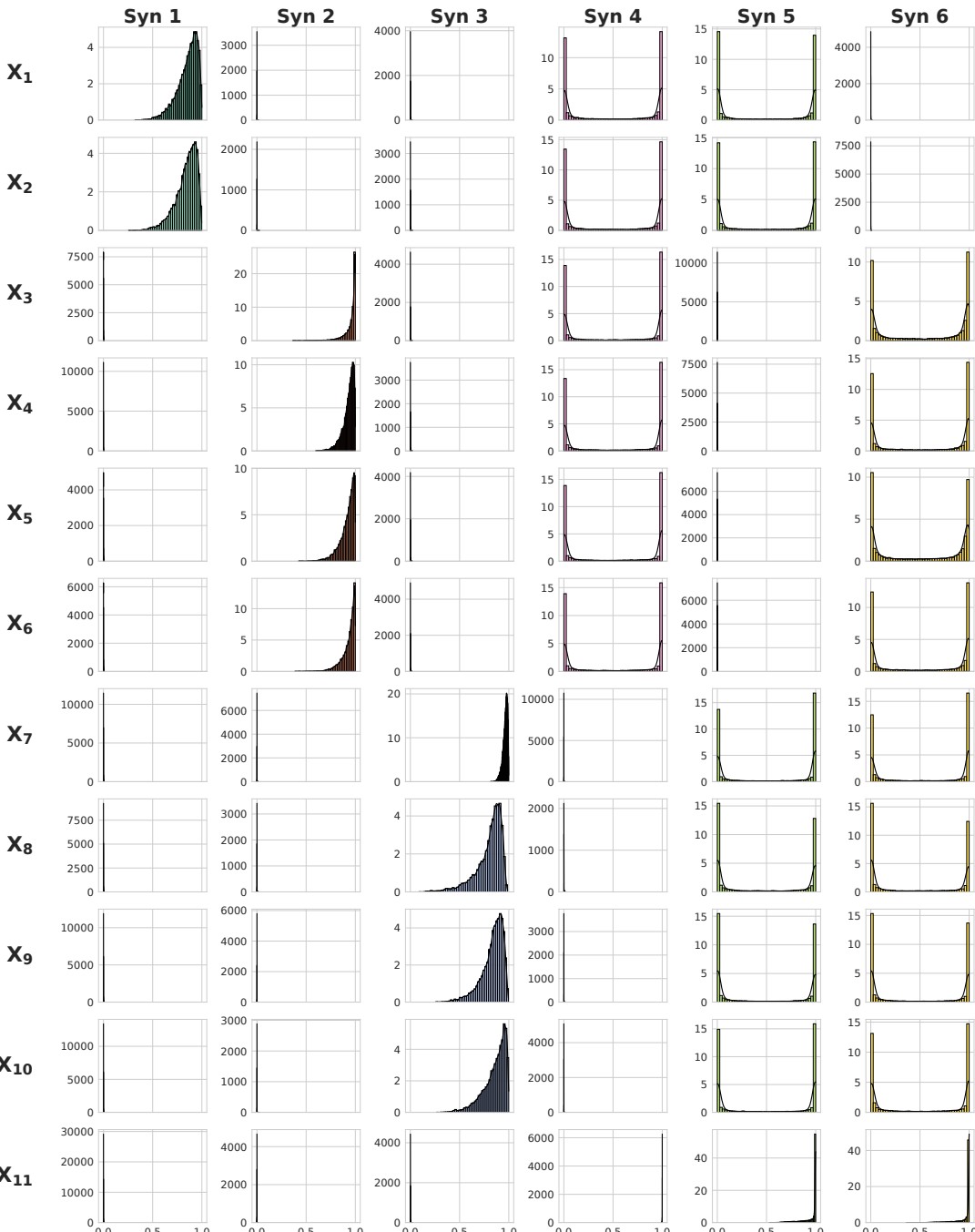

*Figure 8.* The mask distribution for the 6 synthetic experiments in Section 4.1, for one of the 20 experiments. Shown are the histograms and associated KDE plots. $X_{11}$ is the switch-feature, used in Syn4–6.

## D.2. Predictive Performance

Figure 9 shows the predictive performance (AUROC) of the models used in Section 4.1. Hide&Seek has consistently higher predictive performance than comparable selector–predictor models INVASE, L2X and REAL-x, despite having significantly fewer parameters.

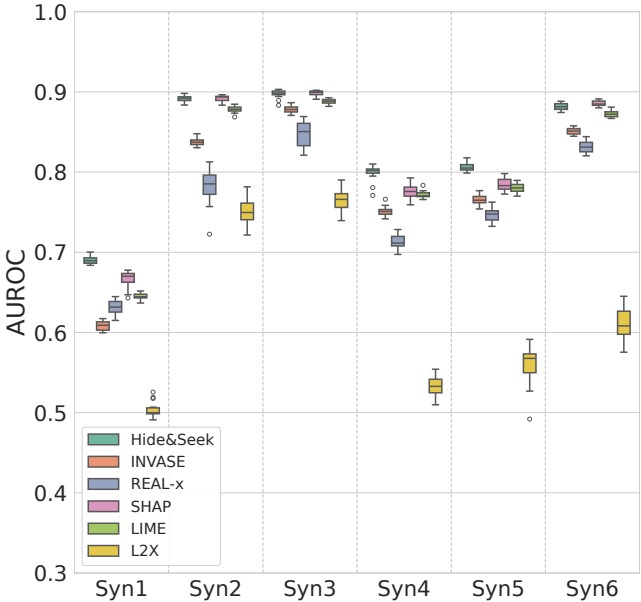

*Figure 9.* Predictive performance (AUROC) of the models in Section 4.1.

The poorer predictive performance of REAL-x is likely due to out-of-distribution draws when predicting on highly parsimonious data (Hooker et al., 2019). Recall that the predictor network in REAL-x is trained disjointly on feature masks drawn from a Bernoulli(0.5) distribution.

## D.3. $\lambda$ Sensitivity

Three of the models: Hide&Seek, INVASE and REAL-x have the parameter $\lambda$ (or $\lambda_{max}$ for Hide&Seek, which we will refer to as $\lambda$, here) which balances the trade-off between parsimony and prediction accuracy. When tuning, AUROC is typically calculated on a validation dataset for varying values of $\lambda$. Another available metric is the percentage significance, which reports the proportion of instance-wise features that the model has deemed important. The goal is to choose a $\lambda$ which provides high parsimony while preserving high prediction accuracy. Figure 10 shows these metrics on a validation dataset for each of the models, for varying values of $\lambda$. It provides a number of useful insights.

Firstly, we see that Hide&Seek has a robust sensitivity to varying values of $\lambda$. While INVASE and REAL-x quickly become too parsimonious as $\lambda$ increases, Hide&Seek maintains a percentage significance close to the ground truth. This is likely due to the annealing schedule, which ramps up the weight of the parsimony term in the loss function towards the end of training. Secondly, Hide&Seek maintains a strong AUROC, with almost all of its AUROC values above 0.8. Conversely, REAL-x quickly loses prediction accuracy as $\lambda$ increases. Recall that the predictor network in REAL-x is trained disjointly on feature masks drawn from a Bernoulli(0.5) distribution. Therefore, when a highly parsimonious feature set is received, REAL-x performance could degrade due to out-of-distribution (OOD) shift (Hooker et al., 2019).

We have also added, post-hoc, the F1 instance-wise feature importance scores to the graphs to demonstrate the relationship between the AUROC used in tuning and the underlying IWFS metric we are trying to optimize. Note that there is a strong correlation between Hide&Seek's AUROC and the IWFS F1 results. Specifically, the AUROC initially increases as parsimony increases, implying that the model's predictions could be benefiting from discarding unimportant features. Conversely, the highest AUROC for REAL-x occurs when it has the most features available. INVASE has instances of high prediction accuracy corresponding to a low F1 score. The graphs of AUPRC vs $\lambda$ displayed similar patterns.

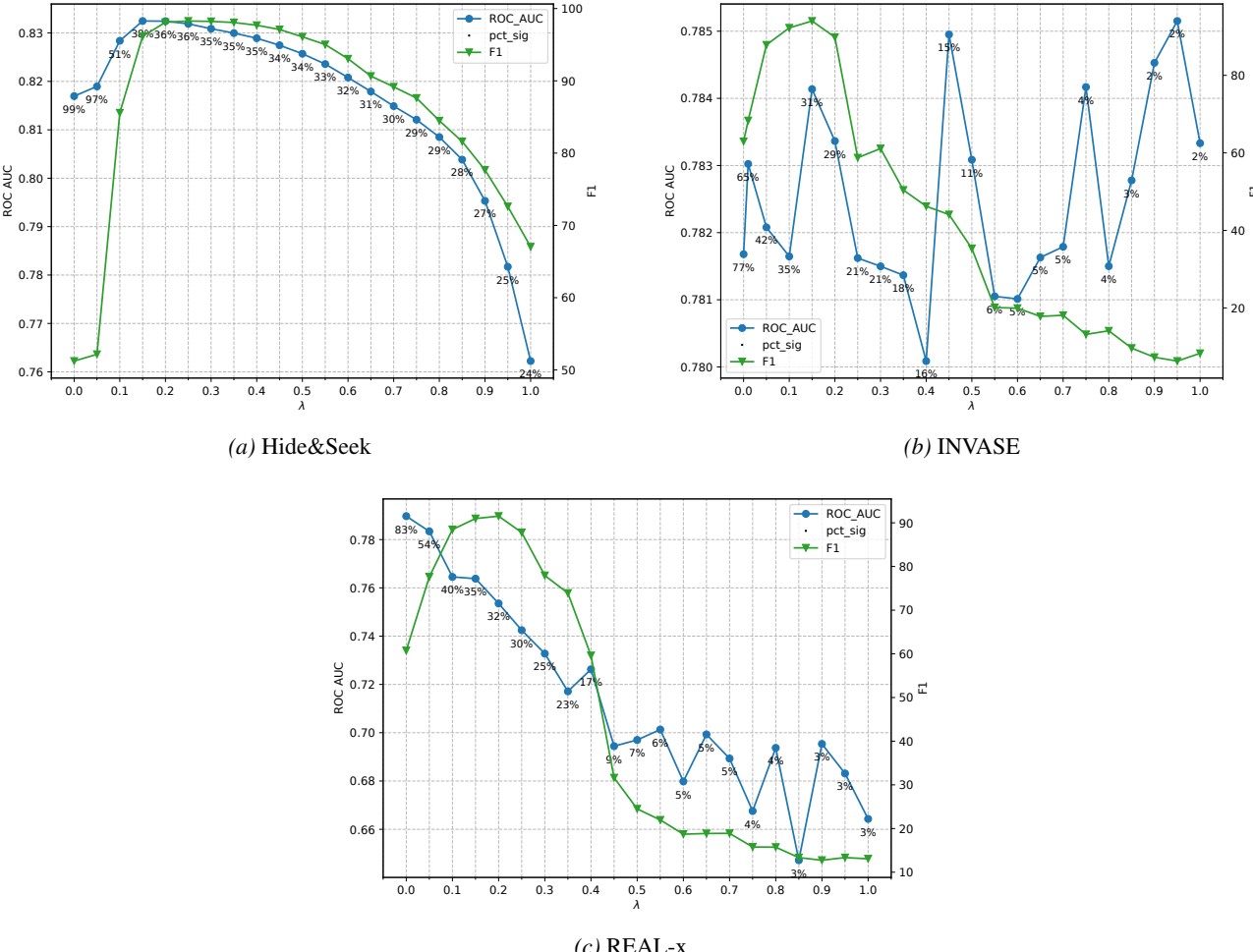

*(a)* Hide&Seek

*(b)* INVASE

*(c)* REAL-x

*Figure 10.* $\lambda$ sensitivity across different models, with AUROC (predicting $Y$) and F1 (identifying important features) averaged over Syn1-6.

The INVASE results reported in Section 4.1 use $\lambda = 0.1$, which is the value used in the original INVASE paper for the same synthetic data experiments. As shown, this corresponds to high IWFS F1 performance. If, instead, the $\lambda$ with highest AUROC had been selected ($\lambda = 0.95$ or $\lambda = 0.45$), the INVASE IWFS results would have been lower.

**D.4. Results for $k = 3$ and $k = 4$**

For SHAP, LIME, L2X, LASSO, and RForest, the top $k$ important features need to be specified. In the experiments of the main text, $k$ is chosen based on the number of ground truth important features sought for each dataset. Specifically, $k = 2$ for Syn1, $k = 4$ for Syn2–3 and $k = 5$ for Syn4–6. This may overestimate the FDR for Syn4 and Syn5, which have only 3 important features when $X_{11} < 0$. To account for this, SHAP, LIME and L2X results for $k = 3$ and $k = 4$ are shown in Table 8.

*Table 8.* TPR and FDR for Syn4–5 for different values of $k$, as explained in Section 4.1. Each metric is the median of 20 experiments.

| Model | k | Syn4 | | Syn5 | |
|---|---|---|---|---|---|
| | | TPR | FDR | TPR | FDR |
| Hide&Seek | | 99 | 4 | 97 | 3 |
| SHAP | 5 | 72 | 38 | 74 | 38 |
| | 4 | 60 | 35 | 63 | 34 |
| | 3 | 46 | 34 | 48 | 32 |
| LIME | 5 | 54 | 51 | 50 | 54 |
| | 4 | 40 | 50 | 41 | 51 |
| | 3 | 30 | 50 | 30 | 50 |
| L2X | 5 | 44 | 65 | 53 | 57 |
| | 4 | 36 | 65 | 45 | 55 |
| | 3 | 27 | 65 | 35 | 53 |

**D.5. Run Time Analysis**

Table 9 reports run times for the instance-wise feature selection models. INVASE's training time is substantially longer than other methods, due to its REINFORCE architecture. REAL-x employs differentiable training using REBAR gradients (Tucker et al., 2017) but requires separate training of the selector and predictor networks. There is also a significant difference in model complexity. INVASE uses $\approx$100k parameters, REAL-x uses $\approx$57k and Hide&Seek uses $\approx$3k. Hide&Seek does not use batching, which is present in REAL-x, INVASE and L2X. See Appendix B for further detail on model designs.

*Table 9.* Typical model run times on the synthetic data, with IWFS results reported in table 2. Times include training (10,000 samples), prediction, and feature attribution (10,000 samples). All models were run on identical hardware, described in Appendix B.1.

| Method | Run time (hh:mm:ss) |
|---|---|
| SHAP | 00:00:02 |
| L2X | 00:00:03 |
| Hide&Seek | 00:00:05 |
| REAL-x | 00:01:16 |
| LIME | 00:10:54 |
| INVASE | 01:18:52 |

**D.6. Syn3 - Specification vs. Implementation**

We note a minor discrepancy between the specification of model Syn3 in the previous works (Yoon et al., 2018; Chen et al., 2018) and the code linked in their publication. For our experiments, we use the data-generating model of the previous code, so that the results are comparable.

*Table 10.* Paper and code expressions for Syn3 model in INVASE, L2X, and Hide&Seek.

| Method | Source | Syn3 |
|---|---|---|
| INVASE | Paper | $-10\sin(2X_7) + 2|X_8| + X_9 + \exp(-X_{10})$ |
| | Code | $-10\sin(0.2X_7) + |X_8| + X_9 + \exp(-X_{10}) - 2.4$ |
| L2X | Paper | $-100\sin(2X_1) + 2|X_2| + X_3 + \exp(-X_4)$ |
| | Code | $-100\sin(0.2X_1) + |X_2| + X_3 + \exp(-X_4) - 2.4$ |
| Hide&Seek | Both | $-10\sin(0.2X_7) + |X_8| + X_9 + \exp(-X_{10}) - 2.4$ |

## D.7. Parsimony-Weight Annealing Analysis

As outlined in section 3.1, the annealing schedule for $\lambda_t$ over $t$ epochs is $\lambda_t = \left(\frac{t}{T}\right)^q \lambda_{\max}$, where $q = 2$ for all our experiments. Table 11 shows the stability of the annealing schedule for different choices of $q$. It includes a mix of instance-wise feature selection (TPR, FDR, F1) and prediction (AUROC) metrics. We see that for square root and linear growth, $q \in \{0.5, 1\}$, the results are poor, as expected. However, for $q \in \{2, 3, 4, 5\}$, the results are stable. This is because the values in the second set allow the model to emphasize prediction accuracy early in training and parsimony later, as demonstrated in Figure 2. As $q$ increases (for a fixed $\lambda_{\max}$), the number of epochs spent in larger values of $\lambda_t$ decreases, resulting in less parsimony.

*Table 11.* Sensitivity analysis for choices of $q$ in $\lambda_t = \left(\frac{t}{T}\right)^q \lambda_{\max}$. TPR, FDR and F1 are instance-wise feature selection metrics and AUROC is a prediction metric. Each value represents the mean across the six synthetic datasets in Section 4.1 and 20 seeds. $\lambda_{\max} = 0.3$ for all runs.

| $q$ | TPR | FDR | F1 | AUROC |
|---|---|---|---|---|
| 0.5 | 38.34 | 5.33 | 44.45 | 0.68 |
| 1.0 | 78.43 | 2.71 | 82.88 | 0.79 |
| 2.0 | 97.66 | 2.49 | 97.26 | 0.83 |
| 3.0 | 98.53 | 4.30 | 96.67 | 0.83 |
| 4.0 | 98.78 | 5.44 | 96.13 | 0.83 |
| 5.0 | 98.88 | 6.65 | 95.44 | 0.83 |

Figure 11 shows the impact of different choices of $q = \{0, 1, 2, 3\}$ on the loss function (12) for different values of $\lambda_{\max}$. Each row shows (a) The cross-entropy term $\sum_{c=1}^{C} y_c \log \hat{y}_c$ vs t. (b) The regularized parsimony term $\frac{\lambda_t}{d} \|\mathbf{m}\|_1$ vs t. (c) The combined loss $\ell(\boldsymbol{\alpha}, \boldsymbol{\beta}, t)$ vs t. The metrics are calculated on a hold-out validation set using the same data as in the Syn4 experiment in section 4.1. Figure 11 shows that once $\lambda_{\max}$ is set large enough to impose parsimony, the choice $q = 2$ provides the best results. It allows Hide&Seek to prioritize prediction accuracy early and then mask parsimony in later training epochs. It is more stable than $q = 3$ and results in a lower final combined loss. Note that when $t = 500$, the loss function has the same value for all four values of $q$ and is therefore comparable.

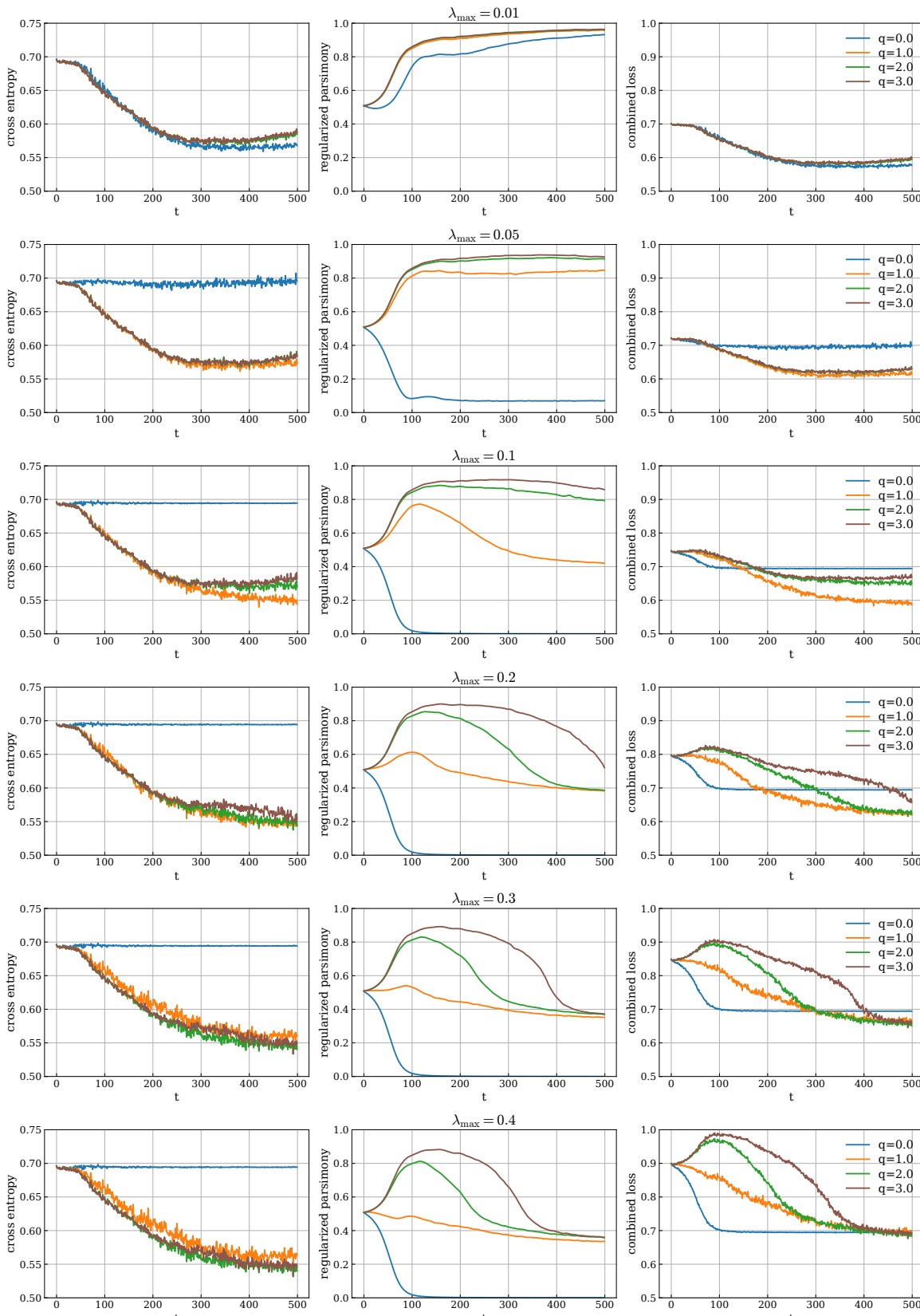

*Figure 11.* Parsimony-weight annealing analysis. These graphs show the impact of different choices of $q = \{0, 1, 2, 3\}$ in $\lambda_t = \left(\frac{t}{T}\right)^q \lambda_{\max}$ on the loss function (12), for different values of $\lambda_{\max}$. Each row shows (a) The cross-entropy term $\sum_{c=1}^{C} y_c \log \hat{y}_c$ vs t. (b) The regularized parsimony term $\frac{\lambda_t}{d} \|\mathbf{m}\|_1$ vs t. (c) The combined loss $\ell(\boldsymbol{\alpha}, \boldsymbol{\beta}, t)$ vs t. See section D.7 for more detail.

# E. Further experiments

This section provides the following experiments: training the models on 1,000,000 samples; performance on 100 features; and California Housing data.

## E.1. Training on 1,000,000 Samples

Table 12 shows the results from a single experiment for each model and dataset, where the training data was 1,000,000 samples. Note that L2X has improved results when trained on more data. Hide&Seek still outperforms other models.

*Table 12.* Performance of IWFS algorithms on Syn1-6 when trained on 1,000,000 samples.

| Model | Hide&Seek | | INVASE | | REAL-x | | SHAP | | LIME | | L2X | |
|---|---|---|---|---|---|---|---|---|---|---|---|---|
| | TPR | FDR | TPR | FDR | TPR | FDR | TPR | FDR | TPR | FDR | TPR | FDR |
| Syn1 | **100** | **0** | **100** | **0** | **100** | **0** | 98 | 2 | 24 | 76 | **100** | **0** |
| Syn2 | **100** | **0** | **100** | **0** | 87 | 6 | **100** | **0** | **100** | **0** | **100** | **0** |
| Syn3 | 99 | **0** | **100** | **0** | 94 | **0** | **100** | **0** | 98 | 2 | 85 | 15 |
| Syn4 | **100** | **1** | 90 | **1** | 95 | 4 | 70 | 38 | 55 | 49 | 83 | 32 |
| Syn5 | **98** | **1** | 84 | **1** | 89 | **1** | 73 | 36 | 50 | 53 | 90 | 29 |
| Syn6 | **98** | **1** | 90 | **1** | 95 | 6 | 76 | 24 | 51 | 49 | 91 | 9 |

Note that unlike Hide&Seek and INVASE, we found that the parsimony regularizer $\lambda$ in REAL-x had to be tuned down as the number of training samples grew from $N = 10,000$ (as in section 4.1) to $N = 1,000,000$.

## E.2. Training on 100 Features and Ensembling

We conduct an experiment in which we increase the number of synthetic features from 11 to 100. The relationship between features remains as described in Section 4.1. There are now an additional 89 noise signals.

Additionally, we introduce Hide&Seek$_{ens}$, an extension of our base architecture that leverages ensembling and column subsampling. Taking advantage of the model's fast training, we train an ensemble of 10 independent models, each observing a random 90% subset of the features. Instance-wise feature importance is then found by averaging the masks across the ensemble and applying the standard $> 0.5$ selection threshold.

We also present two runs of INVASE, with no architectural change, showing the results on two values of $\lambda$. As noted in Section D.3, INVASE is hard to tune and the optimal $\lambda$ might not be easily read off a $\lambda - \text{AUROC}$ tuning curve (see Figure 10b for an example). In Table 13, INVASE corresponds to the logical choice of $\lambda$ during tuning (1.2) while INVASE$_{ideal}$ corresponds to the results if the ideal $\lambda$ (0.6) was known.

In the results, we see that the base Hide&Seek is competitive. Hide&Seek$_{ens}$ outperforms all other models and is as good as, if not better than, the ideal INVASE. These results show the applicability of Hide&Seek to large datasets and outline that ensembling and column subsampling could be useful in improving results.

*Table 13.* F1 values for high-dimensional (100 features) synthetic datasets. The target is the same function of $\{X_1, \ldots, X_{11}\}$ as in the earlier experiments. This experiment includes 89 extra unimportant features of independent noise. Each F1 value is the median of 20 experiments. Hide&Seek$_{ens}$ is an ensemble of 10 independent Hide&Seek models with 90% column subsampling and INVASE$_{ideal}$ represents INVASE's performance if the ideal $\lambda$ is known.

| Model | Hide&Seek | Hide&Seek$_{ens}$ | INVASE | INVASE$_{ideal}$ | REAL-x | SHAP | LIME | L2X |
|---|---|---|---|---|---|---|---|---|
| | F1 | F1 | F1 | F1 | F1 | F1 | F1 | F1 |
| Syn1 | 90 | 100 | 0 | 100 | 48 | 14 | 2 | 2 |
| Syn2 | 96 | 100 | 100 | 100 | 72 | 92 | 100 | 10 |
| Syn3 | 97 | 99 | 100 | 100 | 93 | 88 | 78 | 4 |
| Syn4 | 66 | 73 | 53 | 63 | 68 | 56 | 25 | 4 |
| Syn5 | 77 | 79 | 63 | 75 | 68 | 53 | 25 | 4 |
| Syn6 | 70 | 75 | 84 | 85 | 68 | 72 | 40 | 5 |

### E.3. California Housing

We present a new experiment that evaluates the models against correlated real-world features in a semi-synthetic setting. This experiment includes a switch feature with three partitions. The California Housing dataset contains information on housing block groups in California from the US 1990 Census (Pace & Barry, 1997). Each block group represents an average of 1425.5 people living in proximity. There are 20,640 block groups. We use the following variables: *Median Income, Median House Age, Average Rooms, Average Bedrooms, Population, Occupancy* (average members per household) and *Longitude*.

We establish a ground truth feature importance by sampling $Y$ from a Bernoulli distribution: $P(Y = 1|U) = \frac{1}{1+e^U}$, where the feature importance is based on the switch-feature *Longitude*, with three geographic partitions:

- Where Longitude $< -121.5$: $U = $ Average Rooms $-$ Average Bedrooms

- Where $-121 \leq$ Longitude $< -118$: $U = \frac{\text{Population}}{\text{Occupancy}}$

- Where Longitude $\geq -118$.: $U = 5 \times$ Median Income $- 2 \times$ (Median House Age)$^2$

We split the data into 12,828 training samples, 1,604 validation samples and 1,604 test samples. Features were standardized using the training data. For Hide&Seek, REAL-x and INVASE, we tuned $\lambda$ across [0, 0.05, 0.1, 0.25, 0.5, 0.75, 1, 1.25, 1.5]. We then assessed each model's IWFS performance in recovering the ground truth features (housing attributes and longitude). The results are shown in Table 14.

*Table 14.* IWFS performance on California Housing data. Each TPR, FDR and F1 metric is the median of 10 runs of each model on the same test data.

|  | Hide&Seek | INVASE | REAL-x | SHAP | LIME | L2X |
|---|---|---|---|---|---|---|
| TPR | 97 | 84 | 79 | 74 | 56 | 52 |
| FDR | 17 | 17 | 10 | 26 | 44 | 48 |
| F1 | 88 | 83 | 83 | 74 | 56 | 52 |

## F. Extra detail

This section provides further detail on the non-synthetic data experiments of the paper.

### F.1. Credit Default Data Detail

The features in the Credit default data experiment in Section 4.4 are: LIMIT_BAL, BILL_AMT1, BILL_AMT2, BILL_AMT3, BILL_AMT4, BILL_AMT5, BILL_AMT6, PAY_AMT1, PAY_AMT2, PAY_AMT3, AGE, PAY_AMT4, PAY_AMT5, PAY_AMT6, SEX, EDUCATION, MARRIAGE, PAY_0, PAY_2, PAY_3, PAY_4, PAY_5 and PAY_6. The binary target for the raw data is *default payment next month*, which is not used in Syn4–6. See https://archive.ics.uci.edu/dataset/350/default+of+credit+card+clients for more details. The correlation matrix is shown in Figure 12.

### F.2. MNIST Detail

In the MNIST experiment in section 4.5, the hyperparameter $\lambda$ for Hide&Seek, INVASE and REAL-x was tuned over $\{0.05, 0.1, 0.2, 0.3, 0.4, 0.5\}$. We also compared two scaling methods: global rescaling to the $[0, 1]$ range and feature-wise standardization to zero mean and unit variance (using the training set). The settings associated with the highest prediction accuracy were selected. For Hide&Seek, zero mean and unit-variance scaling resulted in a higher prediction accuracy for all $\lambda$ values. $\lambda = 0.05$ was used, although explanation patches were largely similar for all 6 values.

### F.3. Breast Cancer Subtype Classification Detail

Table 15 contains a list of the 100 genes used in the experiment in section 4.6. Table 6 shows the top 10 genes identified by each model, ranked by their mean importance scores. Table 16 provides more detail inclduing standard errors.

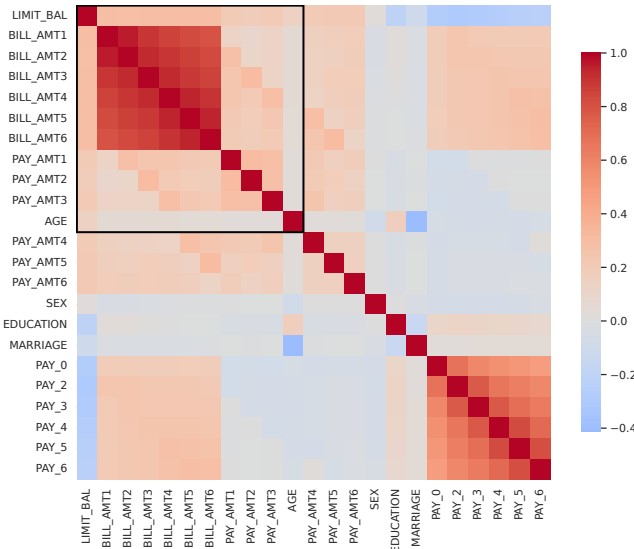

*Figure 12.* The correlation matrix for the 23 input features in the Credit default data experiments. The 11 features surrounded by the black box are used for $\{X_1, \ldots, X_{11}\}$ in generating Syn4–6 in Section 4.4.

*Table 15.* 100 genes used for analysis in section 4.6 to match the same experiment as in Covert et al. (2021).

| 100 genes | | | | | | | | |
|---|---|---|---|---|---|---|---|---|
| OSTbeta | STATH | MAPK10 | PLEKHG5 | ERO1L | ZNF711 | ZNF385 | OR52E8 | SLC5A11 |
| P4HA3 | LHFPL4 | MGC33657 | CAPZB | RBM15B | C1orf176 | KLF3 | OLFM4 | NBR2 |
| CCDC64 | NUP210 | HEMGN | SLC25A3 | LEF1 | MVD | OTUD3 | KIAA1949 | SLC44A3 |
| ZNF775 | THY1 | DYNC1I2 | CYP1A1 | SPTA1 | CLEC4M | RXFP3 | TSHR | C7 |
| CRYBB2 | PPAPDC3 | TXNL4B | CHST9 | HACE1 | AYTL1 | PRSS35 | ZNF408 | DDC |
| CSTL1 | OR2F1 | C12orf50 | SH3YL1 | SNUPN | COL25A1 | HPS4 | ZFPM1 | OAS2 |
| TUBA1C | OR8K5 | THSD3 | ATP6V0C | RAB22A | AP1B1 | CTAGE6 | C6orf26 | ESR1 |
| UPK3B | ROBO4 | TMEFF1 | KIAA1279 | ZFP36L1 | GRINA | YTHDF3 | TMCC1 | UBE1DC1 |
| C6orf15 | PDE6A | PEO1 | TMEM52 | PARP1 | GSS | RDH11 | STXBP1 | ACLY |
| TMSB10 | TUBB | LIPK | HRC | C20orf111 | OMA1 | NCAPH2 | GPX2 | BPY2C |
| ZNF324 | CDC27 | CCNB2 | CNOT7 | BIRC3 | GAL3ST3 | PLEKHM1 | SPOCD1 | PENK |
| TAS2R9 | | | | | | | | |

*Table 16.* Detailed gene importance (mean ± standard error). Importance represents mean mask size (Hide&Seek, INVASE, REAL-x), mean importance scores (SHAP, LIME), or mean selection frequency (L2X).

| Hide&Seek | | INVASE | | REAL-x | |
|---|---|---|---|---|---|
| Gene | Importance ± SEM | Gene | Importance ± SEM | Gene | Importance ± SEM |
| ESR1 | 0.999 ± 0.000 | CCNB2 | 0.723 ± 0.037 | ESR1 | 0.969 ± 0.003 |
| CCNB2 | 0.951 ± 0.014 | ESR1 | 0.711 ± 0.040 | CCNB2 | 0.897 ± 0.006 |
| STATH | 0.827 ± 0.018 | ZNF775 | 0.680 ± 0.039 | NUP210 | 0.833 ± 0.010 |
| C6orf26 | 0.804 ± 0.017 | KLF3 | 0.619 ± 0.038 | C6orf15 | 0.758 ± 0.013 |
| TUBB | 0.784 ± 0.024 | C6orf15 | 0.610 ± 0.040 | SLC25A3 | 0.754 ± 0.010 |
| C7 | 0.773 ± 0.022 | TMSB10 | 0.597 ± 0.038 | SPOCD1 | 0.606 ± 0.011 |
| NCAPH2 | 0.734 ± 0.021 | NCAPH2 | 0.586 ± 0.032 | C6orf26 | 0.593 ± 0.015 |
| UPK3B | 0.727 ± 0.033 | C20orf111 | 0.570 ± 0.036 | TUBB | 0.506 ± 0.015 |
| PARP1 | 0.710 ± 0.036 | NUP210 | 0.562 ± 0.041 | OR52E8 | 0.502 ± 0.019 |
| HACE1 | 0.680 ± 0.031 | CAPZB | 0.553 ± 0.036 | HACE1 | 0.498 ± 0.014 |

| SHAP | | L2X | | LIME | |
|---|---|---|---|---|---|
| Gene | Importance ± SEM | Gene | Importance ± SEM | Gene | Importance ± SEM |
| ESR1 | 0.695 ± 0.029 | PENK | 0.640 ± 0.048 | TUBB | 0.044 ± 0.004 |
| CCNB2 | 0.402 ± 0.016 | BIRC3 | 0.600 ± 0.049 | HACE1 | 0.044 ± 0.005 |
| C6orf15 | 0.212 ± 0.007 | TMEM52 | 0.590 ± 0.049 | C6orf26 | 0.038 ± 0.004 |
| ZNF385 | 0.126 ± 0.007 | HPS4 | 0.590 ± 0.049 | PENK | 0.033 ± 0.004 |
| NUP210 | 0.121 ± 0.009 | OTUD3 | 0.590 ± 0.049 | ESR1 | 0.033 ± 0.004 |
| TMSB10 | 0.077 ± 0.006 | CAPZB | 0.590 ± 0.049 | C7 | 0.032 ± 0.004 |
| C6orf26 | 0.071 ± 0.004 | C6orf26 | 0.580 ± 0.050 | CCNB2 | 0.031 ± 0.003 |
| C7 | 0.069 ± 0.004 | STXBP1 | 0.580 ± 0.050 | KIAA1949 | 0.029 ± 0.003 |
| HACE1 | 0.064 ± 0.006 | ACLY | 0.580 ± 0.050 | NCAPH2 | 0.029 ± 0.004 |
| GPX2 | 0.062 ± 0.004 | COL25A1 | 0.580 ± 0.050 | OAS2 | 0.029 ± 0.003 |

