# OpenReview forum: "Hide&Seek: Learning to Explain in an End-to-End Differentiable Network"
_ICML.cc/2026/Conference — ICML 2026 regular_

### Official Review · Reviewer_Dkvq · 2026-02-24

**Soundness:** 1
**Presentation:** 2
**Significance:** 2
**Originality:** 2
**Overall Recommendation:** 4
**Confidence:** 4

**Summary:**

This paper proposes Hide&Seek, an instance-wise feature selection (IWFS) method that jointly trains a selector (“Hide”) and a predictor (“Seek”) in an end-to-end differentiable framework. The authors provide a theoretical argument that ablating unselected features with a fixed value can leak information, leading to misleading attributions. To mitigate this, they replace each feature partially using a continuous mask, mixing the original feature with a stochastic replacement value, which also keeps training differentiable. Empirically, Hide&Seek achieves stronger explanation quality than prior IWFS baselines such as L2X and INVASE.

**Compliance With Llm Reviewing Policy:**

Affirmed.

**Final Justification:**

This paper proposes an instance-wise feature selection (IWFS) method based on selector and predictor architectures. One of the paper’s strengths is that its feature-ablation design is well motivated: the authors provide evidence that the common choice of constant ablation can lead to feature leakage, and they address this by partially ablating features using a mixture of the original value and values sampled from the marginal distributions. I initially had concerns about the marginal feature-ablation strategy; however, during the rebuttal period, the authors provided additional experiments, including results with an alternative ablation scheme, which helped address part of this concern. Nevertheless, the paper still has some fundamental limitations, most notably that the method cannot be applied to black-box models, even though the reported results are promising. Overall, I am raising my score to 4 from my initial score of 3.

**Key Questions For Authors:**

The paper has merits, but I have a number of concerns and find the current empirical evaluation insufficiently convincing. Please see my questions below, depending on your responses (and any additional evidence you can provide), I will re-evaluate my score and may consider increasing it.

1) Please discuss and compare against [1], which also introduces a “Hide-and-Seek” framework (though with different design choices). Clarify the novelty of your method relative to [1], and include [1] as an additional baseline in the experimental section.

2) Please consider adding more recent baselines, such as LEX [2], to better reflect the current state of the literature.

3) The reporting of experimental results is somewhat inconsistent. In particular, please provide AUROC for the first three synthetic datasets in Table 5, and also report the AUROC of the underlying predictive models used by SHAP and LIME. I am concerned that the weak SHAP/LIME results may partially reflect poor base-model performance, rather than limitations of the explanation method itself. More generally, several results raise concerns about baseline tuning. For example, on Syn3 (100 features), INVASE improves relative to the 11-feature setting, which could indicate that the initial INVASE configuration was not properly tuned. Similarly, the SHAP result on Syn1 with 100 features appears unusually low.

4) Also, please report the standalone predictive performance of the ``Seek'' model (without the selector) to assess whether any accuracy is being sacrificed in favor of explainability.

5) The paper replaces missing features by sampling from their marginal distributions; given the difficulty of conditional sampling, this is a reasonable design choice. However, please clarify how the marginal distributions are estimated in practice. More importantly, the synthetic datasets appear to be generated from i.i.d. Gaussian features, which aligns well with marginal replacement and may inadvertently favor the proposed approach. Given this, the experimental scope feels limited: beyond the synthetic benchmarks, experiments are conducted on only three additional datasets, with no quantitative comparison on MNIST and no baseline comparison at all on the breast cancer dataset. I encourage the authors to expand evaluation on more realistic real-world datasets, including settings with more than 10 classes (or otherwise more complex label structure), and to test multiple architecture families to better assess generalization.

6) The method contains several interacting components, making it difficult to isolate what drives performance; beyond the existing sweep over $\lambda_{\max}$, please include additional ablations that test alternative feature replacement strategies (e.g., constant-value ablation vs.\ marginal sampling) and compare using $z = m \odot x$ alone versus $z = m \odot x + (1-m)\odot \hat{x}$.


References

[1] Tagaris, Thanos, and Andreas Stafylopatis. "Hide-and-seek: A template for explainable AI." arXiv preprint arXiv:2005.00130 (2020).

[2] Senetaire, Hugo Henri Joseph, et al. "Explainability as statistical inference." International Conference on Machine Learning. PMLR, 2023.

**Limitations:**

Please consider discussing the following points as limitations of your work:

1) The method requires joint end-to-end training of the selector and predictor networks, which restricts its use to settings where the model can be trained from scratch. This limits applicability to pre-trained black-box models where only input–output access is available and explanations must be generated post hoc.

2) Feature importance is represented as a continuous mask $m \in [0,1]^d$, and the paper applies a thresholding criterion (e.g., $m > 0.5$) to decide whether a feature is ''selected'', which can be somewhat arbitrary and may be sensitive to calibration and the chosen threshold.

**Strengths And Weaknesses:**

Strengths:

1) The paper tackles an important and practical problem. In particular, Theorem 1 provides a valuable formalization of the information-leakage issue and is a meaningful contribution.

2) The experimental setup is clearly described and the authors release code for reproducibility. However, the SHAP-based baseline implementation appears to be missing from the provided repository.

3) The method achieves strong empirical performance and is reported to train substantially faster than several existing IWFS approaches.

Weaknesses:

1) The approach relies on end-to-end training of a selector–predictor pair, which limits applicability to black-box models where only input–output access is available and explanations must be generated post hoc.

2) While the paper motivates leakage and proposes stochastic replacement, it does not provide a formal argument or guarantee that this replacement strategy prevents information leakage in general (beyond the fixed-ablation counterexample).

3) The paper frames “ablating missing features with a constant value” as a key weakness of prior methods, but this is largely an implementation choice rather than a fundamental limitation. For example, methods such as INVASE or REAL-X could, in principle, be combined with conditional or marginal distribution imputation.

4) Although the empirical results are strong, the evaluation is mostly on relatively small-scale datasets, leaving scalability and effectiveness on larger, more complex settings less clear.

---

> ### Author Rebuttal · Authors · 2026-03-31
>
> We thank reviewer Dkvq for their considered review, particularly identifying Theorem 1's value in explaining information leakage. Thank you also for sharing [8]. As an arXiv preprint we had not come across it, but it is excellent and contains useful parallels. We shall refer to the [8] model as *HnS* below, and plan to cite it.
>
> # Comparison to HnS
>
> **Similarities**
>
> Both HnS and our framework use a selector network to hide features and a predictor network to predict using the adjusted features. Both optimize a loss function balancing prediction accuracy and parsimony, as in [1, 2, 9]. We both discovered that emphasizing prediction accuracy early in training and parsimony later is important, and use an annealing schedule to achieve this.
>
> **Key Differences**
>
> Like INVASE and prior methods, HnS uses 0-ablation to remove unselected features. As outlined in our paper, this can lead to information leakage, a key motivation for our model. We take an entirely different approach: our masks do not discretely select or reject features, but replace a proportion of each feature with marginal noise. This is unique compared to prior amortized explanation methods [1, 2, 8, 9].
>
> Unlike HnS (and [1, 9]), our feature replacement strategy is directly differentiable. Like [9], HnS defaults to a *reinforce* strategy. While INVASE relies on computationally expensive reinforcement learning, HnS accelerates this by condensing the REINFORCE penalty into a single backpropagation step (eq 16 in [8]).
>
> Another difference is our parsimony weight annealing schedule. Quadratically growing our $\lambda_t$ during training (section 3.1) enables stable exploration of the loss landscape (Figure 2), resulting in convergence for both parsimony and prediction terms. Experimentally, this approach markedly improved results.
>
> As requested, we implemented the HnS framework for our tabular setting using stochastic thresholding, the default *reinforce* estimator, and a pre-trained Seeker network:
>
> ### *Table: [https://github.com/anonymous1861/hide_and_seek/blob/main/images/HnS%20comparison.png](https://github.com/anonymous1861/hide_and_seek/blob/main/images/HnS%20comparison.png)*
>
> # Improved AUROC reporting
>
> Per the reviewer's request, we provide the AUROC for Syn1-3 for the credit default data. Initially, we omitted Syn1-3 because Syn4-6 represent harder problems involving a switch-feature. However, upon reflection, these provide valuable information and will be included.
>
> ### *Table: https://github.com/anonymous1861/hide_and_seek/blob/main/images/AUROC_update.png*
>
> You also requested the AUROC results for the underlying models. As suspected, the underlying SHAP model had not been adequately tuned. The LIME model, however, was fine. We have since updated our SHAP model for the Synthetic data and present the AUROC results in the figure below, which will be included in our paper.
>
> ### *Figure: https://github.com/anonymous1861/hide_and_seek/blob/main/images/auroc_models.png*
>
> Updated SHAP TPR/FDR results (minor):
>
> ### *Table: https://github.com/anonymous1861/hide_and_seek/blob/main/images/SHAP_update.png*
>
> Increasing the underlying model's complexity somewhat reduced SHAP's ability to identify switch-features. The MNIST SHAP model is unaffected, already having 99%+ prediction accuracy. The L2X model has no hyper-parameters to tune.
>
> As noted in [8], our goal is not pushing the boundary of prediction accuracy, but finding *good enough* predictions to enable truthful explanations. This XAI aspect is our primary motivation.
>
> # Notes
>
> - We attempted to implement LEX [6] as a baseline. Their code was unlinked in the paper, though we found this repo: [https://github.com/HugoSenetaire/LEX/tree/main](https://github.com/HugoSenetaire/LEX/tree/main). Unfortunately, missing modules and broken dependencies prevented implementation. Please note we provide a comparison to Knockoffs in our response to reviewer mQpq.
>
> - We will discuss Hide\&Seek's applicability to pre-trained black-boxes in our limitations section. Note that the LEX paper [6] discusses this (Post-Hoc vs In-Situ explanations) and ascribes benefit to our setting.
>
> - Regarding point 6: INVASE provides the constant value ablation comparison. Constant value ablation does fit our framework as we replace a proportion of each feature. We tested $z = m \odot x$ early on and found it ineffective. This is logical, as $z = m \odot x$ alone simply scales down the feature's magnitude (in other methods, it acts as a selection probability). In our setting, the tension between $m$ and $1-m$ trades off between *truth* and *noise*, allowing the model to improve prediction accuracy while increasing parsimony.
>
> - The LIME AUROC indicates the standalone Seek module's performance, as they have similar underlying infrastructure. We can run the full experiment given time.
>
> - See our response to reviewer gCnN for an analysis of the masking threshold.
>
> - Reference list is found with reviewer mQpq

---

> > ### Author Rebuttal · Reviewer_Dkvq · 2026-04-03
> >
> > I thank the authors for their responses. However, one of my main concerns remains insufficiently addressed, namely the relationship between marginal distribution sampling and the formulation of the synthetic datasets. In addition, the number of real-world datasets is still limited, and the HnS baseline results are reported only on the synthetic datasets. Due to these remaining concerns, I will keep my score unchanged at 3 for now.
> >
> >
> > Edit on April 7:
> >
> > I thank the authors for their responses. Based on the new results, I am raising my score to 4. In the revised version, please include all additional experiments conducted during the rebuttal period, and also discuss the reasons for the poor performance of Knockoff sampling on the Syn2 dataset.

---

> > > ### Author Response · Authors · 2026-04-07
> > >
> > > Thank you for your thoughtful comments.
> > >
> > > To address your considerations we present:
> > > 1. A comparison to conditional sampling, using knockoffs for IWFS, for the correlated synthetic datasets
> > > 2. An application of ensembling to our algorithm for better performance on correlated data
> > > 3. Another experiment using california housing data.
> > >
> > > ## Correlated data analysis, including Knockoff (conditional) comparison, switch-analysis and ensembling
> > > We explored Model-X knockoffs [10] as an option for feature replacement in Equation (3) of the paper, as an alternative conditional feature replacement strategy. Unlike an earlier response to another reviewer, this Knockoff usage is completely instance-wise and plugs directly into our framework (the replacement remains proportional) as an alternative for marginal replacement.
> > >
> > > Mathematically, Model-X knockoffs guarantee that the joint data distribution remains invariant when swapping a fixed subset of features $S$, expressed as:
> > >
> > > $$(X, \tilde{X})_{\text{swap}(S)} \stackrel{d}{=} (X, \tilde{X})$$
> > >
> > > Note, however, that instance-wise architectures determine the masked subset dynamically using a Selector network, meaning the chosen features are a function of the data itself, $S = f(X)$. In this case:
> > >
> > > $$(X, \tilde{X})_{\text{swap}(f(X))} \not\stackrel{d}{=} (X, \tilde{X})$$
> > >
> > > Nonetheless, we present it below as an alternative method for marginal replacement - one that is closer to the true conditional distribution.
> > >
> > > Our results indicate that while IWFS knockoffs perform similarly to marginal sampling on standard synthetic data, they offer no improvement on semi-synthetic, correlated data.
> > >
> > > Furthermore, the primary flaw in marginal sampling would be the model learning from out-of-distribution values, leading to information leakage, as outlined in the paper. To investigate this, we tested the switch-accuracy ($X_{11}$) in the credit-default data and found that marginal sampling maintains **100% switch accuracy** across all settings.
> > >
> > > Our analysis suggests that the correlation challenge is not driven by information leakage from an out of distribution replacement strategy, but rather by the common challenge of features sharing importance, which typically results in lower TPR and higher FDR.
> > >
> > > To address this, we leveraged the fast training time of our framework to ensemble multiple models. We implemented a voting scheme across 10 Hide\&Seek models using column subsampling (0.9) and marginal replacement. As shown in the results, marginal sampling does not lead to information leakage, and performance on correlated data is significantly improved through this ensembling approach.
> > >
> > > ### *Table: https://github.com/anonymous1861/hide_and_seek/blob/main/images/F1_IWFS_knockoff_ensemble.png*
> > >
> > > ## California Housing Experiment
> > >
> > > The California Housing dataset contains information on housing block groups in California from the US 1990 Census. Each block group represents an average of 1425.5 people living in proximity. There are 20,640 block groups and the following variables: *Median Income, Median House Age, Average Rooms, Average Bedrooms, Population, Occupancy* (average members per household), *Latitude* and *Longitude*. We use this dataset to examine the robustness of our model in the presence of correlated features and with more than two switch conditions.
> > >
> > > We established a ground truth feature importance by sampling $Y$ from a Bernoulli distribution
> > >
> > > $$P(Y=1 | X) = 1/(1+e^{-X})$$
> > >
> > > where the predictive signal $X$ is determined by three geographic switch conditions:
> > >
> > > * For Longitude < -121.5: $X = \text{Average Rooms} - \text{Average Bedrooms}$
> > > * For -121.5 $\leq$ Longitude < -118.5: $X = \text{Population} / \text{Occupancy}$
> > > * For Longitude $\geq$ -118.5: $X = 5 \times \text{Median Income} - 2 \times (\text{Median House Age})^2$
> > >
> > > We split the data into 16,512 training samples, 2,064 validation samples and 2,064 test samples. Outliers were removed from the training data, leaving 12,824 for training. All features were standardized to the training data.
> > >
> > > We then assessed each model's IWFS to recover the ground truth features (housing attributes and longitude). The results are shown:
> > >
> > > ### Table: https://github.com/anonymous1861/hide_and_seek/blob/main/images/cali_housing_performance.png
> > >
> > >
> > > ### References
> > > [10] Romano, Y., Sesia, M., \& Candès, E. (2020). Deep knockoffs. Journal of the American Statistical Association, 115(532), 1861–1872.

---

### Official Review · Reviewer_mQpq · 2026-03-02

**Soundness:** 2
**Presentation:** 3
**Significance:** 2
**Originality:** 2
**Overall Recommendation:** 3
**Confidence:** 5

**Summary:**

The authors propose a novel dynamic feature selection algorithm that, instead of removing features, propose a balancing mask between the real feature and a generated one, using the same distribution found in the dataset. The experimental results over synthetic datasets are promising.

**Compliance With Llm Reviewing Policy:**

Affirmed.

**Final Justification:**

The rebuttal improves the clarity of the experimental section and partially addresses my concerns. However, I remain unconvinced that the empirical validation is sufficient to support the paper’s claims. In my initial review, I specifically asked for results on the full MNIST dataset, as this is a standard benchmark used by prior instance-wise feature selection methods. The authors chose to report results only on a restricted subset (digits 3 vs. 8), and the rebuttal does not provide the requested full-dataset evaluation.

This limitation makes it difficult to properly assess the competitiveness and generality of the proposed method relative to existing work. While I appreciate the authors’ efforts in the rebuttal, the experimental section still feels incomplete, and for this reason I am maintaining my initial score.

**Key Questions For Authors:**

How your approach differs from the Knockoff papers?

How the model behaves against the state-of-the-art in both MNIST and the Breast Cancer dataset?

**Limitations:**

yes

**Strengths And Weaknesses:**

- **Soundness:**
    + (+) The idea is interesting, easy to reproduce and to implement
    + (-) I would like to see an ablation study regarding the effect of the network in the results. The Appendix suggests the authors used different network architectures per model.
    + (-) The experimental section is very limited. It only shows a comparison against other methods in the synthetic data. Only image cues are provided for MNIST, and the importance of each selected variable for the Breast dataset.

- **Presentation:**
    + (+) The paper is easy to reproduce and to implement
    + (-) Information regarding the experiments configuration is only accessible in the Appendix.

- **Significance:**
    + (+) Better dynamic feature selection methods are always welcomed.
    + (-) The lack of experimental results does not allow to measure the effect of this contribution.

- **Originality:**
    + The authors claim that they draw $\mathbf{\hat{x}}$ from the product of the marginal distributions, as in Quantitative Input Influence. Although not completely exact, this is similar to the approach used in the Knockoff models [1], which were also applied in dynamic feature selection [2]. This field of study is not mentioned nor referenced in the paper. This contribution seems rather similar to those approaches.

[1] Lu, Y., Fan, Y., Lv, J., & Stafford Noble, W. (2018). DeepPINK: reproducible feature selection in deep neural networks. Advances in neural information processing systems, 31.

[2] Paul, D., Bardhan, S., Saha, S., & Mathew, J. (2023). ML-KnockoffGAN: Deep online feature selection for multi-label learning. Knowledge-Based Systems, 271, 110548.

---

> ### Author Rebuttal · Authors · 2026-03-31
>
> We thank reviewer mQpq for their considered review.
>
> # Knockoffs
>
> The primary reason for not including Knockoffs as a benchmark is that they have traditionally motivated global, not instance-wise, feature selection. This is the case in both papers the reviewer has shared [3] and [4]. Our work is concerned with instance-wise feature selection (IWFS). IWFS assigns an importance score to each instance (e.g. patient or student) in a dataset. This has potential use beyond standard global feature selection. For example, in identifying individual genetic influences in patient outcomes or developing policy for student subpopulations.
>
> Knockoffs were also compared against INVASE in [9] and the results there (see Table 1) show that they do not perform as well as other algorithms (INVASE, SHAP, LIME). As we were comparing ourselves to these algorithms, and had already included two global feature selection algorithms, we determined not to explore Knockoffs.
>
> As requested, we ran our synthetic data experiment on Knockoffs as well. We used Gaussian model-X knockoffs https://web.stanford.edu/group/candes/deep-knockoffs/ and compared a Lasso evaluator with Deep Pink [3]. Hide\&Seek has superior performance.
>
> ### *Table https://github.com/anonymous1861/hide_and_seek/blob/main/images/knockoff_comparison.png*
>
> The table shows median TPR and FDR from 20 experiments, per dataset, per model. These are directly comparable with Table 2 of our paper.
>
> A pathway for future work could be to use Knockoffs to generate $\hat{x}$ in equation 3 of our paper. As such, we plan to now reference Knockoffs in section 2 and outline it as a possible area of future work at the end of our paper.
>
> Despite possible future explorations, we believe this work offers substantial standalone value. We explain a clear failure mode in existing selector-predictor networks. We introduce a feature-replacement strategy fundamentally different from prior approaches [1, 2, 8, 9]. Our strategy is the only directly differentiable method yet presented in amortized explanations. It leads to exceptionally fast performance and achieves 100\% switch feature identification.
>
> # TCGA Breast Cancer experiments
>
> Following the request of the reviewer, we have extended our TCGA Breast Cancer experiment to include a comparison with the other state-of-the-art models. We did not initially do this as the TCGA data represents a $C=4$ classification problem, whereas the [1, 2, 9] papers and code only considered binary classification. We have now updated their code (which we will make available in our GitHub repo) to handle more than 2 classes.
>
> The results show that INVASE, REAL-x and SHAP also identify ESR1 and CCNB2 as the most important genes, while LIME and L2X struggle. Recall that ESR1 [5] and CCNB2 [7] were found to be important in the literature. Hide\&Seek and SHAP provide a meaningful gap between the first two genes and the others. This experiment provides further validation for Hide\&Seek's performance on a large number of features (100) and correlated data.
>
> ### *Table: https://github.com/anonymous1861/hide_and_seek/blob/main/images/tcga_model_comparison.png*
>
> # Notes
> - We did not use different network architectures per model. This is outlined in our appendix. The only difference between experiments was the tuning of the hyperparameter $\lambda_{max}$, similarly to [2, 9].
>
> Bibliography for all reviewers:
>
> [1] Jianbo Chen et al. Learning to explain: An information-theoretic perspective on model interpretation. In International Conference on Machine Learning. PMLR, 2018.
>
> [2] Neil Jethani et al. Have we learned to explain?: How interpretability methods can learn to encode predictions in their interpretations. In International Conference on Artificial Intelligence and Statistics. PMLR, 2021.
>
> [3] Yang Lu et al. DeepPINK: Reproducible feature selection in deep neural networks. In Advances in Neural Information Processing Systems, volume 31, 2018.
>
> [4] Dipanjyoti Paul et al. ML-KnockoffGAN: Deep online feature selection for multi-label learning. Knowledge-Based Systems, 271, 2023.
>
> [5] Dan R. Robinson et al. Activating ESR1 mutations in hormone-resistant metastatic breast cancer. Nature Genetics, 45(12), 2013.
>
> [6] Hugo Henri Joseph Senetaire et al. Explainability as statistical inference. In International Conference on Machine Learning. PMLR, 2023.
>
> [7] Emman Shubbar et al. Elevated cyclin b2 expression in invasive breast carcinoma is associated with unfavorable clinical outcome. BMC Cancer, 13(1), 2013.
>
> [8] Thanos Tagaris and Andreas Stafylopatis. Hide-and-seek: A template for explainable ai. arXiv preprint arXiv:2005.00130, 2020.
>
> [9] Jinsung Yoon et al. Invase: Instance-wise variable selection using neural networks. In International conference on learning representations, 2018.

---

> > ### Author Rebuttal · Reviewer_mQpq · 2026-04-03
> >
> > I am still not very convinced about the differences with the knockoff theory, but I am open to raise my score if the other if the authors provide independence on the backbone model used. More experiments are needed in other to support the paper claims.

---

> > > ### Author Response · Authors · 2026-04-07
> > >
> > > Thank you for your thoughtful comments. Here we provide:
> > > 1. Further consideration of. Knockoffs
> > > 2. Analysis on correlated data and ensembling of our model
> > > 3. A new experiment on California Housing data.
> > >
> > > ## Exploration of Knockoffs for Feature Replacement
> > > We explored Model-X knockoffs [10] as an option for feature replacement in Equation (3) of the paper, as an alternative conditional feature replacement strategy. Unlike our earlier run, this Knockoff usage is completely instance-wise and plugs directly into our framework (the replacement remains proportional) as an alternative for marginal replacement.
> > >
> > > Mathematically, Model-X knockoffs guarantee that the joint data distribution remains invariant when swapping a fixed subset of features $S$, expressed as:
> > >
> > > $$(X, \tilde{X})_{\text{swap}(S)} \stackrel{d}{=} (X, \tilde{X})$$
> > >
> > > Note, however, that instance-wise architectures determine the masked subset dynamically using a Selector network, meaning the chosen features are a function of the data itself, $S = f(X)$. In this case the replacement values are no longer guaranteed to follow the joint distribution:
> > >
> > > $$(X, \tilde{X})_{\text{swap}(f(X))} \not\stackrel{d}{=} (X, \tilde{X})$$
> > >
> > > Nonetheless, we present it as an alternative method for marginal replacement - one that is closer to the true conditional distribution.
> > >
> > > ## Correlated Data Analysis
> > > Our results indicate that while IWFS knockoffs perform similarly to marginal sampling on standard synthetic data, they offer no improvement on semi-synthetic, correlated data.
> > >
> > > Furthermore, the primary flaw in marginal sampling would be the model learning from out-of-distribution values, leading to information leakage, as outlined in the paper. To investigate this, we tested the switch-accuracy ($X_{11}$) in the credit-default data and found that marginal sampling maintains **100% switch accuracy** across all settings.
> > >
> > > Our analysis suggests that the correlation challenge is not driven by information leakage from an out of distribution replacement strategy, but rather by the common challenge of features sharing importance, which typically results in lower TPR and higher FDR.
> > >
> > > To address this, we leveraged the fast training time of our framework to ensemble multiple models. We implemented a voting scheme across 10 Hide\&Seek models using column subsampling (0.9) and marginal replacement. As shown in the results, marginal sampling does not lead to information leakage, and performance on correlated data is significantly improved through this ensembling approach.
> > >
> > > ### *Table: https://github.com/anonymous1861/hide_and_seek/blob/main/images/F1_IWFS_knockoff_ensemble.png*
> > >
> > >
> > > ## California Housing Experiment
> > >
> > > The California Housing dataset contains information on housing block groups in California from the US 1990 Census. Each block group represents an average of 1425.5 people living in proximity. There are 20,640 block groups and the following variables: *Median Income, Median House Age, Average Rooms, Average Bedrooms, Population, Occupancy* (average members per household), *Latitude* and *Longitude*. We use this dataset to examine the robustness of our model in the presence of correlated features and with more than two switch conditions.
> > >
> > > We established a ground truth feature importance by sampling $Y$ from a Bernoulli distribution
> > >
> > > $$P(Y=1 | X) = 1/(1+e^{-X})$$
> > >
> > > where the predictive signal $X$ is determined by three geographic switch conditions:
> > >
> > > * For Longitude < -121.5: $X = \text{Average Rooms} - \text{Average Bedrooms}$
> > > * For -121.5 $\leq$ Longitude < -118.5: $X = \text{Population} / \text{Occupancy}$
> > > * For Longitude $\geq$ -118.5: $X = 5 \times \text{Median Income} - 2 \times (\text{Median House Age})^2$
> > >
> > >
> > > We split the data into 16,512 training samples, 2,064 validation samples and 2,064 test samples. Outliers were removed from the training data, leaving 12,824 for training. All features were standardized to the training data.
> > >
> > > We then assessed each model's IWFS to recover the ground truth features (housing attributes and longitude). The results are shown:
> > >
> > > ### Table: https://github.com/anonymous1861/hide_and_seek/blob/main/images/cali_housing_performance.png
> > >
> > > ### References
> > > [10] Romano, Y., Sesia, M., \& Candès, E. (2020). Deep knockoffs. Journal of the American Statistical Association, 115(532), 1861–1872.

---

### Official Review · Reviewer_d5LG · 2026-03-07

**Soundness:** 3
**Presentation:** 3
**Significance:** 2
**Originality:** 2
**Overall Recommendation:** 3
**Confidence:** 4

**Summary:**

This article's central objective is to present Hide&Seek, an end-to-end differentiable framework for instance-wise feature selection. The core task is to dynamically identify important features for each individual data instance.

An important concept explored by the paper is information leakage. Previous methods replace unimportant features with fixed values, such as zero. This creates a flaw where the replaced value acts as a hidden signal for conditional branching, causing models to miss crucial switch features. Furthermore, their discrete selection processes are non-differentiable and computationally inefficient. To solve these issues, Hide&Seek replaces features by continuously blending them with noise drawn from their marginal distributions. This creates a fully differentiable model that structurally prevents information leakage, while utilizing parsimony-weight annealing to stabilize the optimization process.

In its experiments, Hide&Seek achieved state-of-the-art accuracy on synthetic datasets. It completely eliminated information leakage by successfully identifying 100% of the conditional switch features—which previous models missed approximately 50% of the time. The differentiable architecture also drastically reduced training time from over an hour to just five seconds. Furthermore, the model successfully identified highly correlated features in credit default data, extracted context-specific patches distinguishing digits in MNIST images, and discovered critical, biologically relevant biomarkers like ESR1 in breast cancer microarray data.

**Compliance With Llm Reviewing Policy:**

Affirmed.

**Final Justification:**

The rebuttal partially addressed my concerns but core issues remain, reinforcing my prior assessment. I raised four weaknesses (W1–W4):

**W1 (Originality — limited differentiation from prior work)**: The authors highlighted the direct differentiability of their proportional replacement strategy and showed that Hide&Seek is the only algorithm achieving >95% TPR & <5% FDR across all datasets (Table 2). I accept this differentiation to some extent. **Partially addressed.**

**W2 (Soundness — no formal guarantee that stochastic replacement prevents leakage)**: The switch-feature experiment empirically showed no leakage, and the authors provided intuitive reasoning. However, this remains an empirical argument, not a formal guarantee. The promised discussion in limitations/future work is welcome but insufficient. **Partially addressed.**

**W3 (Originality — fixed-value ablation is an implementation choice, not a fundamental limitation of prior frameworks)**: **Unresolved.** Prior methods like INVASE could in principle also use marginal/conditional draws; framing fixed-value ablation as a fundamental limitation of prior work overstates the novelty. The authors acknowledged the point but did not revise the framing, instead pointing to end-to-end differentiability and speed as practical advantages.

**W4 (Significance — evaluation limited to small-scale datasets)**: The California Housing experiment and Knockoffs comparison are welcome additions. However, core evaluation remains synthetic-data-centric, and large-scale real-world benchmarks are still missing. **Partially addressed.**

**Clarity**: Well-structured and easy to follow.

With W3 unresolved and W2/W4 only partially addressed, I maintain my score of 3 (Weak reject).

**Key Questions For Authors:**

- During inference, how are stochastic replacement values handled? If a random value is selected in the same manner as during training, there is a concern that prediction results could vary significantly depending on the randomly selected value.

**Limitations:**

The limitations of the work were not discussed adequately. Although the potential negative societal impact was discussed, how the negative impact can be addressed was not discussed.

**Strengths And Weaknesses:**

## Strengths
- The proposed model is an architecture that combines a feature selection network with a prediction network, and the authors propose a method to train them simultaneously. This approach is reasonable as it has been observed in existing research. Furthermore, the method where the feature selection network replaces features by blending them with noise is a rational design choice for preventing information leakage. Additionally, annealing the penalty term is an appropriate technique for enhancing training stability.
- The paper is very clearly written with good organization. In the experiments, the behavior of the proposed method is clarified using synthetic data, and then its utility is evaluated on real data, making the results trustworthy.

## Weaknesses
- The Hide&Seek idea is based on the existing approach of combining a feature selection network with a prediction network. It differs from prior work in that (1) the Selector network generates continuous-valued masks, and (2) masked dimensions are replaced with random noise rather than fixed values like zero or mean. However, these are not entirely novel concepts within the XAI context.
- While the paper proposes a method to address the Information Leakage problem and demonstrates significant improvements in the experimental setup, further validation is needed regarding the general utility of the proposed method and its applicability to other tasks and datasets. In particular, it remains unclear whether similar improvements would be achieved on large-scale real-world datasets and more complex tasks.
- In the experiments, zero replacement is performed as a naive feature replacement method, but replacement with data mean values is not conducted.

---

> ### Author Rebuttal · Authors · 2026-03-31
>
> We thank reviewer d5LG for their considered review, in particular their identifying many strengths of our work.
>
> # Originality
>
> The reviewer is correct that a number of instance-wise feature importance methods exist. We fall into a class of models called amortized explanation methods, such as INVASE, REAL-x and L2X, which use a selector model to hide features and a predictor model to make predictions.
>
> Despite the existence of other methods, instance-wise feature selection (IWFS) is by no means solved. Neural networks largely remain black boxes. The need for effective XAI algorithms is pressing. Table 2 of our paper shows that many popular IWFS algorithms have sub-par performance on common benchmarks. We are the only algorithm to achieve >95% TPR and <5% FDR on all six datasets. The next best algorithm, INVASE, is slow to train and its design is flawed by information leakage.
>
> However, the main contributions of our paper are not in the results, but in the novel approach we take to get there. We explain a clear failure mode in existing selector-predictor networks and provide a theoretical underpinning for it with Theorem 1. We introduce a feature-replacement strategy fundamentally different from the existing amortized explanation methods [1, 2, 8, 9], where we replace a proportion of each feature. Our strategy is the only directly differentiable method yet presented in this class.
>
> # Stochastic replacement during inference
>
> We draw the proportional replacement values in prediction in the same way as in training. Currently, these are from the product of marginals, but future improvements could include from the conditional. Specifically, we independently shuffle each column of our entire training set to form $\hat{X}$, from which we take replacement values.
>
> In our main experiments, we performed 20 simulations upon each synthetic dataset and found strong consistency across runs. This addresses your concern that prediction results could vary.
>
> In particular, here are the AUROC results for each of the models, showing that Hide\&Seek has minimal variance between runs.
>
> ### Figure: https://github.com/anonymous1861/hide_and_seek/blob/main/images/auroc_models.png
>
> Figure 5 in our appendix shows boxplots for TPR, FDR and F1 scores. There, Hide\&Seek has one of the lowest variances in explanation outcomes.
>
> # Limitations
>
> We will update our limitations section to reflect key ideas raised by some reviewers.
>
> # Notes
> We note a key challenge in the development of IWFS algorithms, namely, that ground truth feature importance is rarely known and there is dearth of rigorous benchmark datasets. In this context many papers rely on synthetic experiments. We have supplemented this with semi-synthetic credit default data, MNIST experiments and TCGA Breast cancer data. Following reviewer feedback, we have expanded our TCGA Breast cancer data experiment to include comparison to other models.
>
> The results show that INVASE, REAL-x and SHAP also identify ESR1 and CCNB2 as the most important genes, while LIME and L2X struggle. Recall that ESR1 [5] and CCNB2 [7] were found to be important in the literature. Hide&Seek and SHAP provide a meaningful gap between the first two genes and the others. This experiment provides further validation for Hide&Seek's performance on a large number of features (100).
>
> ### *Table: https://github.com/anonymous1861/hide_and_seek/blob/main/images/tcga_model_comparison.png*
>
> We believe that our paper contributes meaningfully to the field. We deepen the understanding of information leakage, we present an effective feature replacement strategy not yet seen in amortized explanation methods and we raise the benchmark on performance and efficiency.
>
> - Reference list is found with reviewer mQpq

---

> > ### Author Rebuttal · Reviewer_d5LG · 2026-04-02
> >
> > I thank the authors for the detailed rebuttal and the effort to provide additional experiments.
> >
> > However, two concerns remain:
> >
> > First, W3 is not directly addressed. My point was not that z = m ⊙ x fails, but that prior methods such as INVASE could in principle also replace ablated features with marginal or conditional draws rather than fixed values — making the "fixed-value ablation" framing a critique of their implementation choice, not a fundamental limitation of their framework. I would appreciate the authors' direct response to this distinction.
> >
> > Second, regarding W4, the additional experiments (TCGA extension with 572 samples, Knockoff comparison on synthetic data, masking threshold analysis on synthetic data) remain small-scale and do not constitute validation on large-scale real-world datasets or more complex tasks. My original concern about scalability and generalizability to larger, more realistic settings is not resolved.
> >
> > I maintain my score.

---

> > > ### Author Response · Authors · 2026-04-07
> > >
> > > Thank you for your thoughtful comments.
> > >
> > > In regards to W3. Your point is relevant and well received. At this stage, such comparisons are outside the scope of this work. However, we would point to two strengths of our framework: namely our end-to-end differentiability and the subsequent speed of our model. Our proportional sampling avoids continuous relaxation techniques that other methods use, such as Gumbel–Softmax (INVASE) and REBAR gradients (REAL-x). As such, even were we to use marginal sampling or conditional draws under their framing, their methods would still not be end-to-end differentiable and would be less efficient.
> > >
> > > In regards to W4, we present the below analyses:
> > >
> > > ## Knockoffs and Ensembling for Correlated Data
> > > We explored Model-X knockoffs [10] as an option for feature replacement in Equation (3) of the paper, as an alternative conditional feature replacement strategy. Unlike an earlier response to another reviewer, this Knockoff usage is completely instance-wise and plugs directly into our framework (the replacement remains proportional) as an alternative for marginal replacement.
> > >
> > > Our results indicate that while IWFS knockoffs perform similarly to marginal sampling on standard synthetic data, they offer no improvement on semi-synthetic, correlated data.
> > >
> > > Furthermore, the primary flaw in marginal sampling would be the model learning from out-of-distribution values, leading to information leakage, as outlined in the paper. To investigate this, we tested the switch-accuracy ($X_{11}$) in the credit-default data and found that marginal sampling maintains **100% switch accuracy** across all settings.
> > >
> > > Our analysis suggests that the correlation challenge is not driven by information leakage from an out of distribution replacement strategy, but rather by the common challenge of features sharing importance, which typically results in lower TPR and higher FDR.
> > >
> > > To address this, we leveraged the fast training time of our framework to ensemble multiple models. We implemented a voting scheme across 10 Hide\&Seek models using column subsampling (0.9) and marginal replacement. As shown in the results, marginal sampling does not lead to information leakage, and performance on correlated data is significantly improved through this ensembling approach.
> > >
> > > ### *Table: https://github.com/anonymous1861/hide_and_seek/blob/main/images/F1_IWFS_knockoff_ensemble.png*
> > >
> > > ## Addressing 100 Features via Ensembling
> > > We further applied the 10x ensembling and column subsampling (0.9) to the 100-feature dataset. The performance is now competitive with INVASE. Note that due to time constraints, while the INVASE results are the medians of 20 runs (as in the paper), the Hide\&Seek results are from a single run.
> > >
> > > ### *Table: https://github.com/anonymous1861/hide_and_seek/blob/main/images/100_features_ensemble.png*
> > >
> > > ## California Housing Experiment
> > >
> > > The California Housing dataset contains information on housing block groups in California from the US 1990 Census. Each block group represents an average of 1425.5 people living in proximity. There are 20,640 block groups and the following variables: *Median Income, Median House Age, Average Rooms, Average Bedrooms, Population, Occupancy* (average members per household), *Latitude* and *Longitude*. We use this dataset to examine the robustness of our model in the presence of correlated features and with more than two switch conditions.
> > >
> > > We established a ground truth feature importance by sampling $Y$ from a Bernoulli distribution
> > >
> > > $$P(Y=1 | X) = 1/(1+e^{-X})$$
> > >
> > > where the predictive signal $X$ is determined by three geographic switch conditions:
> > >
> > > * For Longitude < -121.5: $X = \text{Average Rooms} - \text{Average Bedrooms}$
> > > * For -121.5 $\leq$ Longitude < -118.5: $X = \text{Population} / \text{Occupancy}$
> > > * For Longitude $\geq$ -118.5: $X = 5 \times \text{Median Income} - 2 \times (\text{Median House Age})^2$
> > >
> > > We split the data into 16,512 training samples, 2,064 validation samples and 2,064 test samples. Outliers were removed from the training data, leaving 12,824 for training. All features were standardized to the training data.
> > >
> > > We then assessed each model's IWFS to recover the ground truth features (housing attributes and longitude). The results are shown:
> > >
> > > ### Table: https://github.com/anonymous1861/hide_and_seek/blob/main/images/cali_housing_performance.png
> > >
> > > ### References
> > > [10] Romano, Y., Sesia, M., \& Candès, E. (2020). Deep knockoffs. Journal of the American Statistical Association, 115(532), 1861–1872.

---

### Official Review · Reviewer_gCnN · 2026-03-11

**Soundness:** 3
**Presentation:** 3
**Significance:** 3
**Originality:** 3
**Overall Recommendation:** 4
**Confidence:** 3

**Summary:**

This paper studies instance-wise feature selection in selector-predictor models and targets the information-leakage issue caused by deterministic feature ablation. It proposes Hide&Seek, an end-to-end differentiable framework that replaces a proportion of each feature with samples from its distribution, together with a parsimony-weight annealing schedule. The evaluation spans synthetic benchmarks, switch-feature analysis, a semi-synthetic credit-default setting with correlated features, MNIST explanations, and a breast-cancer gene-ranking case study.

**Compliance With Llm Reviewing Policy:**

Affirmed.

**Final Justification:**

The core idea is sound and clearly presented, with a concrete failure mode in deterministic ablation and a coherent proportional-masking remedy. My main concerns were dependence/correlation under marginal replacement, the limited systematic correlated-tabular evidence, and missing sensitivity analysis for replacement and annealing (beyond a threshold discussion). The first rebuttal strengthened the mask-threshold justification and the TCGA comparison against multiple baselines. The second rebuttal addresses the remaining items more directly: an instance-wise Model-X knockoffs variant is derived and compared as a closer-to-conditional alternative to marginal replacement; a switch-accuracy diagnostic on credit-default data shows marginal sampling does not in fact induce out-of-distribution leakage, reframing the correlated-data difficulty as shared-importance rather than information leakage; a 10x ensembling + column-subsampling scheme is introduced and shown to substantially improve correlated-data performance and to make the 100-feature setting competitive with INVASE; and an annealing-schedule sensitivity table over six synthetic datasets quantifies stability across exponent choices. Some caveats remain (the 100-feature ensemble result is from a single run versus INVASE medians, and the conditional-replacement story is partly conceptual), but the second rebuttal materially resolves the practical-soundness questions and reinforces and modestly strengthens my prior weak-accept view.

**Key Questions For Authors:**

How sensitive are the results to the choice of replacement distribution for masked features? A convincing answer showing robustness, or a stronger conditional replacement scheme, would increase my confidence in the method’s soundness on correlated data.

Can you provide a more systematic analysis of correlated-feature settings beyond the current semi-synthetic credit-default experiment? Stronger evidence here would improve my assessment of practical significance.

How sensitive is performance to the mask threshold and the annealing schedule? If the method is robust to these choices, my confidence in reproducibility would increase.

Can you compare against stronger conditional-masking or explanation baselines, or explain why the current baseline set is sufficient? A broader comparison could strengthen the significance/originality case.

**Limitations:**

Partially. The paper acknowledges that misleading explanations are possible, but the limitations discussion should more explicitly cover distribution shift induced by marginal replacement, dependence/correlation issues, scaling to higher-dimensional inputs, and the risk of over-trusting instance-wise explanations in high-stakes domains.

**Strengths And Weaknesses:**

The main strength is that the paper identifies a concrete failure mode in prior selector-predictor methods and offers a simple, technically coherent remedy. The contribution is clearly articulated, and the empirical section is reasonably broad: the synthetic benchmarks are strong, the switch-feature analysis reports 100% median switch accuracy for Hide&Seek on Syn4-6, and runtime is much lower than INVASE/REAL-x. The paper also includes credit-default, MNIST, and BRCA analyses, which help demonstrate breadth beyond toy settings.

I also found the presentation generally clear. The problem formulation is easy to follow, the selector/predictor architecture is straightforward, and the connection between information leakage and the switch-feature setting is well motivated. The annealing mechanism is sensible and appears empirically useful.

My main concern is that the masking scheme replaces features using samples from a product of marginals, which can create unrealistic combinations when features are strongly dependent. This matters because one of the practical motivations is explanation under correlated, real-world tabular data. The real-data evidence is also somewhat limited: the credit-default task is semi-synthetic, the MNIST section is qualitative, and the BRCA section is more of a case study than a rigorous comparative benchmark.

I would also like stronger sensitivity analysis for the replacement distribution, the mask threshold, and the annealing schedule, as well as broader comparison to conditional-masking or attribution-style baselines. Finally, Appendix results suggest some sensitivity in higher-dimensional settings (e.g., the 100-feature synthetic experiment), which weakens the “drop-in robust” impression of the method.

---

> ### Author Rebuttal · Authors · 2026-03-31
>
> # Replacement distribution for masked features
>
> The author raises a valid point, which points to an area for future research. Namely, that the replacement for $\hat{x}$ would ideally be drawn from the joint distribution of all features, rather than the product of the marginals. There is a theoretical risk that the Seek network learns which features have been replaced, leading to information leakage. Our methodology reduces this risk through two mechanisms:
>
> 1. Unlike INVASE, REAL-x and L2X, we do not replace with a constant value. As shown in our switch feature experiment, those methods lead to information leakage, while ours does not. To enable leakage in the earlier models, the Seek module simply has to learn the significance of a single constant value. However, in our set up, it would have to first learn the joint distribution of the original data, then learn to recognize when individual features are out-of-distribution. This is a much harder problem. Recall, that in our setting, for a significant proportion of the training, the Seek module has an emphasis on prediction accuracy. Our annealing schedule introduces a ramp up in parsimony towards the end.
>
> 2. Also note that the replacement value $\hat{x}$ does not replace the entire feature, but rather a proportion of it. This means a proportion of the original joint distribution may be retained, making it harder to identify out-of-distribution draws.
>
> Nonetheless, the reviewer raises a valid point. We plan to include a discussion of this in section 2 (Problem Formulation), presenting the conditional draw as ideal and drawing $\hat{x}$ from the product of the marginals as simply one approach. We will also address this in the limitations section at the end of the paper and in suggestions for future work. Considering this, we propose that our paper stands as a valuable contribution for the reasons identified by the reviewer and for its unique proportional replacement strategy among existing amortized explanation methods [1, 2, 8, 9].
>
> Under our current setting, we draw from the product of the marginals by shuffling the columns of our training set each epoch, with replacement.
>
> # Masking threshold
>
> This reviewer and reviewer Dkvq asked about the masking threshold, which is currently set at 0.5. We recall that our masks are continuous, between 0 and 1. Each mask $m_{i,j}$ determines the proportional replacement for each feature $j$, per instance $i$. $m_{i,j}$ is the proportion retained and $1-m_{i,j}$ is the proportion replaced.  In the Figure below, we present the distribution of masks for each feature in our 6 synthetic experiments. Note the close alignment with the data-generating rules defined in section 4.1 of the paper. This shows that a wide range of masking thresholds (close to 0 and 1) would remain effective for determining feature inclusion. For a few features ($X_1, X_2 \text{ in Syn1 and } X_7,\dots X_{10} \text{ in Syn3}$), a lower threshold would produce strong results but a higher threshold would begin to exclude important features. This analysis indicates that lowering the mask threshold may be of benefit. We will certainly include this analysis in the final paper.
>
> ### *Figure: https://github.com/anonymous1861/hide_and_seek/blob/main/images/mask_distribution.png*
>
> # TCGA Breast Cancer experiments
>
> We have extended our TCGA Breast Cancer experiment to include a comparison with the other state-of-the-art models. We did not initially do this as the TCGA data represents a $C=4$ classification problem, whereas the [1, 2, 9] papers and code only considered binary classification. We have updated their code (which we will make available in our GitHub repo) to handle more than 2 classes.
>
> The results show that INVASE, REAL-x and SHAP also identify ESR1 and CCNB2 as the most important genes, while LIME and L2X struggle. Recall that ESR1 [5] and CCNB2 [7] were found to be important in the literature. Hide\&Seek and SHAP provide a meaningful gap between the first two genes and the others. This experiment provides further validation for Hide\&Seek's performance on a large number of features (100).
>
> ### *Table: https://github.com/anonymous1861/hide_and_seek/blob/main/images/tcga_model_comparison.png*
>
> # Limitations and future work
>
> We will update our limitations section to reflect the ideas you have raised, specifically conditional replacement strategies. We strongly believe that our work adds to and meaningfully advances this field. The strong performance and speed of our model could make it a reliable component of larger ensembles to help tackle very high dimensional data, similarly to column subsampling in XGBoost implementations.
>
> - Reference list is found with reviewer mQpq

---

> > ### Author Rebuttal · Reviewer_gCnN · 2026-04-03
> >
> > The authors respond substantively on two of my Key Questions. On the mask threshold, they present per-feature mask distributions across all six synthetic benchmarks showing close alignment with the data-generating rules of Section 4.1, and argue that a wide range of thresholds would effectively separate included from excluded features. This is informative, though a formal sweep of the threshold hyperparameter with reported metrics would have been stronger. On comparative baselines, they extend the TCGA breast cancer study to include INVASE, REAL-x, SHAP, LIME, and L2X after adapting the prior code for multi-class classification, and report that Hide&Seek and SHAP both identify the literature-supported genes (ESR1 and CCNB2) with meaningful separation from lower-ranked features. This directly addresses my request for a stronger comparative benchmark on real genomics data.
> > However, the remaining Key Questions are only partly answered. On sensitivity to the replacement distribution, which was my primary concern, the authors provide a thoughtful theoretical discussion explaining why marginal replacement is harder to exploit than constant-value ablation and why proportional masking partially retains the original joint structure. They commit to expanding the problem formulation and limitations sections accordingly. However, they do not report any empirical ablation comparing alternative replacement mechanisms or a conditional replacement scheme, which I had specifically requested.
> > On correlated-feature settings beyond the semi-synthetic credit-default experiment, no new controlled correlation benchmark is added. The TCGA extension is welcome as a multi-method comparison on real data but does not substitute for a systematic stress test with known correlation structure. On the annealing schedule, the reply mentions it only in passing and does not report stability across different schedule choices. On the higher-dimensional fragility noted in the appendix for the 100-feature synthetic setting, there is no new analysis.
> > The authors commit to expanding limitations to cover conditional replacement strategies. I appreciate this, but the scope I listed (marginal-induced distribution shift, dependence and correlation, scaling to higher dimensions, and caution in high-stakes domains) should be more fully addressed in the revision.

---

> > > ### Author Response · Authors · 2026-04-07
> > >
> > > Thank you for your thoughtful comments.
> > >
> > > ## Exploration of Knockoffs for Feature Replacement
> > > We explored Model-X knockoffs [10] as an option for feature replacement in Equation (3) of the paper, as an alternative conditional feature replacement strategy. Unlike an earlier response to another reviewer, this Knockoff usage is completely instance-wise and plugs directly into our framework (the replacement remains proportional) as an alternative for marginal replacement.
> > >
> > > Mathematically, Model-X knockoffs guarantee that the joint data distribution remains invariant when swapping a fixed subset of features $S$, expressed as:
> > >
> > > $$(X, \tilde{X})_{\text{swap}(S)} \stackrel{d}{=} (X, \tilde{X})$$
> > >
> > > Note, however, that instance-wise architectures determine the masked subset dynamically using a Selector network, meaning the chosen features are a function of the data itself, $S = f(X)$. In this case:
> > >
> > > $$(X, \tilde{X})_{\text{swap}(f(X))} \not\stackrel{d}{=} (X, \tilde{X})$$
> > >
> > > Nonetheless, we present it below as an alternative method for marginal replacement - one that is closer to the true conditional distribution.
> > >
> > > ### Correlated Data Analysis
> > > Our results indicate that while IWFS knockoffs perform similarly to marginal sampling on standard synthetic data, they offer no improvement on semi-synthetic, correlated data.
> > >
> > > Furthermore, the primary flaw in marginal sampling would be the model learning from out-of-distribution values, leading to information leakage, as outlined in the paper. To investigate this, we tested the switch-accuracy ($X_{11}$) in the credit-default data and found that marginal sampling maintains **100% switch accuracy** across all settings.
> > >
> > > Our analysis suggests that the correlation challenge is not driven by information leakage from an out of distribution replacement strategy, but rather by the common challenge of features sharing importance, which typically results in lower TPR and higher FDR.
> > >
> > > To address this, we leveraged the fast training time of our framework to ensemble multiple models. We implemented a voting scheme across 10 Hide\&Seek models using column subsampling (0.9) and marginal replacement. As shown in the results, marginal sampling does not lead to information leakage, and performance on correlated data is significantly improved through this ensembling approach.
> > >
> > > ### *Table: https://github.com/anonymous1861/hide_and_seek/blob/main/images/F1_IWFS_knockoff_ensemble.png*
> > >
> > > ## Addressing 100 Features via Ensembling
> > > We further applied the 10x ensembling and column subsampling (0.9) to the 100-feature dataset. The performance is now competitive with INVASE. Note that due to time constraints, while the INVASE results are the medians of 20 runs (as in the paper), the Hide\&Seek results are from a single run.
> > >
> > > ### *Table: https://github.com/anonymous1861/hide_and_seek/blob/main/images/100_features_ensemble.png*
> > >
> > > ## Annealing stability
> > >
> > > Here we show the stability of the annealing schedule to different choices of $q$ in in $\lambda_{t} = \left(\frac{t}{T} \right)^{q}\lambda_{\text{max}}$. We see that for square root or linear growth, the results are poor, as expected. However for $q \in \\{ 2,3,4,5 \\}$, which all emphasize prediction accuracy early and parsimony late, the results are stable. The table below shows the mean TPR, FDR, F1 (IFWS metrics) and ROC AUC (prediction metrics) for the six synthetic datasets.
> > >
> > > https://github.com/anonymous1861/hide_and_seek/blob/main/images/annealing_stability.png
> > > ***
> > >
> > > ### References
> > > [10] Romano, Y., Sesia, M., \& Candès, E. (2020). Deep knockoffs. Journal of the American Statistical Association, 115(532), 1861–1872.

---

### Decision · Program_Chairs · 2026-04-30

**Decision:**

Accept (regular)

**Comment:**

Reviewers generally found the core idea well motivated, and the rebuttal helped clarify the paper’s contribution through additional analyses on leakage, correlated features, and robustness to design choices. However, the reviewers also noted concerns regarding the lack of formal guarantees, the novelty relative to prior ablation-based methods, and the limited extent of large-scale real-world validation.